



# The New MISR Research Aerosol Retrieval Algorithm: A Multi-Angle, Multi-Spectral, Bounded-Variable Least Squares Retrieval of Aerosol Particle Properties over Both Land and Water

James A. Limbacher[1,2,3], Ralph A. Kahn[1], and Jaehwa Lee[1,4]

[1]Earth Science Division, NASA Goddard Space Flight Center, Greenbelt, 20771, USA
[2]Science Systems and Applications Inc., Lanham, 20706, USA
[3]Department of Meteorology and Atmospheric Science, The Pennsylvania State University, State College, 16802, USA
[4]University of Maryland, College Park, MD, USA

*Correspondence to*: James A. Limbacher (James.Limbacher@nasa.gov)

**Abstract.** Launched in December 1999, NASA's Multi-angle Imaging SpectroRadiometer (MISR) has given researchers the ability to observe the Earth from nine different views for the last 22 years. Among the many advancements that have since resulted from the launch of MISR is progress in the retrieval of aerosols from passive space-based remote-sensing. The MISR operational standard aerosol retrieval algorithm (SA) has been refined several times over the last twenty years,

resulting in significant improvements to spatial resolution (now 4.4 km) and aerosol particle properties. However, the MISR SA still suffers from large biases in retrieved aerosol optical depth (AOD) as aerosol loading increases. Here, we present a new MISR research aerosol retrieval algorithm (RA) that utilizes over-land surface reflectance data from the Multi-Angle Implementation of Atmospheric Correction (MAIAC) to address these biases. This new over-land/over-water algorithm produces a self-consistent aerosol/surface retrieval when aerosol loading is low (AOD < 1); this is combined with a

prescribed surface algorithm using a bounded-variable least squares solver when aerosol loading is elevated (AOD>2). The two algorithms (prescribed + retrieved surface) are then merged as part of our combined-surface retrieval algorithm. Results are compared with AErosol RObotic NETwork (AERONET) validation sun-photometer direct-sun + almucantar inversion retrievals.

Over-land, with AERONET AOD (550 nm) direct-sun observations as the standard, the root-mean squared error

(RMSE) of the MISR RA combined retrieval (n=9680) is ~0.09, with a correlation coefficient (r) of ~0.93 and expected error of ± (0.225*[MISR AOD] + 0.025). For MISR RA-retrieved AOD > 0.5 (n=565), we report Ångström exponent (ANG) RMSE of ~0.36, with a correlation coefficient of ~0.85. Retrievals of ANG and aerosol particle properties such as fine-mode fraction (FMF) and single-scattering albedo (SSA) improve as retrieved AOD increases. For AOD >1.5 (n=45), FMF RMSE is <0.09 with correlation >0.95, and SSA RMSE is <0.02 with a correlation coefficient >0.80.

Over-water, comparing AERONET AOD to the MISR RA combined retrieval (n=4590), MISR RA RMSE is ~0.06 and r is ~0.94, with an expected error of ±(0.20*[MISR AOD] + 0.01). ANG sensitivity is excellent when MISR RA reported AOD > 0.5 (n=211), with a RMSE of 0.30 and r=0.88. Due to a lack of coincidences with AOD >1 (n=20), our



conclusions about MISR RA high-AOD particle property retrievals over water are less robust (FMF RMSE=0.12 and r=0.96, whereas SSA RMSE=0.022 and r=0.32).

It is clear from the results presented that the new MISR RA has excellent sensitivity to aerosol particle properties (including FMF and SSA) when retrieved AOD exceeds 1-1.5, with qualitative sensitivity to aerosol type at lower AOD, while also eliminating the AOD bias found in the MISR SA at higher AODs. These results also demonstrate the advantage of using a prescribed surface when aerosol loading is elevated.

# 1 Introduction

The first of three Along Track Scanning Radiometer (ATSR) instruments was launched in July 1991, bringing to the attention of the research community some of what multi-angle remote sensing offers (e.g., Flowerdew & Haigh, 1995; North et al., 1999). As NASA began to develop its Earth Observing System in the late 1980s, it also chose to pursue a multi-angle imaging approach by selecting the Multi-angle Imaging SpectroRadiometer (MISR) as one of five instruments to be launched on its flagship Terra spacecraft. MISR was designed to image Earth's surface and atmosphere at nine angles (70.5°, 60.0°, 45.6°, 26.1° in the forward and aft directions along the flight path, plus nadir), in each of four wavelengths (centered at 446, 558, 672, and 866 nm; Diner et al., 1998). Beginning in February 2000, MISR has since acquired more than two decades of approximately once-weekly, global data.

The initial concept for the MISR aerosol and over-land surface retrieval algorithm was developed by Diner and Martonchik (1984a; 1984b; 1985). The method is inherently multi-angle; it assumes that aerosol amount and properties are constant over a retrieval region and uses empirical orthogonal functions (EOFs) in view angle to characterize the directional surface reflectance contributions to the top-of-atmosphere reflectance. Implementation of this approach in the operational MISR Standard Aerosol retrieval algorithm (SA) is described by Martonchik et al., (1998; 2002; 2009). Substantial advances to the SA involved adding a separate process that assumes the shape of the surface angular reflectance is independent of wavelength (Diner et al., 2005) and reducing the size of the retrieval regions from 17.6 km to 4.4 km (Garay et al., 2020). Still, even with the upgrades described above, the MISR SA continues to show a significant negative bias in AOD when aerosol loading is elevated (Kahn et al., 2005; 2010, Kahn and Gaitley, 2015). In addition to this bias in AOD, it is also likely that SA-retrieved aerosol particle properties are negatively impacted at high AODs over-land, as errors in the retrieved surface reflectance will likely manifest themselves as errors in both AOD and aerosol type.

Among most EOS-era satellite imagers, aerosol property information is a unique contribution the MISR instrument can make. As such, a Research Aerosol retrieval algorithm (RA) was developed in parallel with the SA, focused primarily on deriving as much information as possible about particle microphysical properties. This means the RA includes a broader range of particle optical model options in the algorithm climatology than the MISR SA. It results in more subtle particle property distinctions under favorable retrieval conditions, for example, in smoke and volcanic plumes, when the AOD is



sufficiently high (e.g., Flower & Kahn, 2020; Junghenn Noyes et al., 2020). However, especially at low AOD, when particle type discrimination is poorer, having a larger algorithm particle-type climatology can increase AOD uncertainty.

Previously, in the RA, the surface was characterized either by Fresnel-reflecting dark water with possible whitecaps and under-light contributions, or by a more complex surface specified from external sources (Kahn et al., 2001;
Chen et al., 2008). The MISR RA has also provided validation and suggested upgrades to the SA. Initial sensitivity studies established that three-to-five bins in particle size, two-to-four bins in particle single-scattering albedo (SSA), and spherical vs. randomly oriented non-spherical particle properties could be distinguished from MISR data, provided the mid-visible aerosol optical depth (AOD) exceeds about 0.15 or 0.2 (Kahn et al., 1997; 1998, 2001; Kalashnikova & Kahn, 2006). A high bias in retrieved low-AOD values, along with limitations in the MISR radiometric calibration, the algorithm climatology of
particle optical models, and the surface assumptions in these early algorithms (Kahn et al., 2010) were subsequently addressed. The advances initially focused on over-water retrievals. They included modernizing the code, allowing for regional coverage with pixel-level (1.1 km) retrievals, improving the particle optical models, along with better pixel selection, cloud screening and uncertainty assessment (Limbacher & Kahn, 2014). The MISR radiometric calibration applied in the RA was revised based on empirical image analysis, aimed primarily at improving sensitivity to particle properties
(Limbacher & Kahn, 2015). Further refinements included self-consistently retrieving aerosol and Chlorophyll-a over a dark ocean surface, further refining the MISR radiometric calibration to account for temporal degradation (Limbacher & Kahn, 2017), and extending these retrievals to deriving spectral surface albedo for shallow, turbid, and eutrophic water under a Lambertian surface-reflectance assumption (Limbacher & Kahn, 2019).

The current paper takes a further step in the advance of the MISR RA, incorporating over-land aerosol retrievals
with the surface optical model either retrieved self-consistently within the algorithm or prescribed from the MODerate resolution Imaging Spectroradiometer (MODIS) Multi-Angle Implementation of Atmospheric Correction (MAIAC) product (Lyapustin et al., 2018, *Lyapustin and Wang,* 2018). MAIAC accumulates MODIS observations over eight days to produce multi-angle data for the surface retrieval and reports the bi-directional reflectance distribution function (BRDF) at 1 km horizontal resolution. The current paper is organized as follows: Section 2 describes the RA over-land and over-water
retrieval algorithms in detail, for both the prescribed and retrieved surfaces. It introduces the Bounded-Variable Least Squares (BVLS) approach adopted for the prescribed surface version of the algorithm, a new retrieved-surface aerosol retrieval algorithm (over both land and water), and modifications to the particle optical model climatology and other differences from earlier RA versions. The aerosol quantities reported here are AOD at 550 nm, fine-mode AOD fraction at 550 nm, fine-mode effective radius, SSA, and SSA spectral slope ("Brown Smoke" AOD fraction), and coarse-mode non-
spherical AOD fraction (*Junghenn Noyes et al*., 2020). Section 3 presents the results: detailed validation of the over-land and over-water MISR RA retrievals against coincident AERONET sun photometer data/inversions. Conclusions are given in Section 4.


## 2 Methodology

### 2.1 MISR RA General Description

The current MISR RA, presented in this paper, is essentially composed of two sets of retrieval algorithms, both of which derive aerosol loading and properties at 1.1 km resolution: the retrieved-surface algorithm retrieves the Lambertian water-leaving radiance (over water) and applies a spectrally invariant angular-shape-similarity assumption to derive the surface reflectance (over land) [*Diner et al.*, 2005], whereas the other algorithm prescribes the surface reflectance for both land and water from other sources.

Top-of-atmosphere (TOA) reflectances are computed from the MISR radiance data according to the following:

$$\rho_{\lambda,c}^{\text{TOA}} = L_{\lambda,c} * \frac{\pi * D^2}{E_\lambda^{TOA}}, \tag{1}$$

where $L_{\lambda,c}$ represents the observed TOA radiance (W m$^{-2}$ μm$^{-1}$ sr$^{-1}$) in band $\lambda$ and camera c, D is the Earth-Sun distance at the time of observation in Astronomical Units (AU), and $E_\lambda^{TOA}$ is the exo-atmospheric solar irradiance at 1 AU (W m$^{-2}$ μm$^{-1}$). We then correct these TOA reflectances for the following: gas absorption, out-of-band light, stray-light from instrumental artifacts, flat-fielding, and temporal calibration trends [*Limbacher and Kahn,* 2015; 2017; 2019]. Once the TOA reflectances have been corrected for these artifacts, MODIS-MAIAC surface reflectance BRDF kernels [*Lyapustin et al.*, 2018, *Lyapustin and Wang,* 2018] are interpolated temporally to the MISR overpass date. These MAIAC data and the corresponding MISR data are then gridded to a static grid identical for each orbit at the native MISR 1.1 km resolution. Additionally, we interpolate MISR's digital elevation model (DEM) from the MISR ancillary geographic product (AGP) to the 1.1 km grid. To create the validation dataset used in the current paper, gridding in performed instead at 1 km resolution, on a 48x48 pixel box centered on each AERONET site and ingested into the RA. Over land, where MAIAC BRDF kernels are available, the algorithm then converts MAIAC BRDF kernels to surface reflectance for each of MISR's 36 channels, adjusting to ensure that the surface reflectance at any angle never exceeds 3 times the albedo (for a given band) or drops below 33% of the albedo for a given band. Over water, the prescribed surface reflectance is assumed to be Lambertian ([0.0257, 0.00668, 0.00093, 0.0000635] for the blue, green, red, and NIR) once glint is subtracted [*Limbacher & Kahn*, 2017]. The algorithm then runs both sets of retrievals for each scene, one with a prescribed surface (using MAIAC over land and a fixed surface reflectance over water), and one where the surface reflectance is retrieved. Using a newly created land/water mask derived from the MISR retrieved surface algorithm itself, we then consolidate the output (AOD, aerosol properties, cost function, etc.) from the four (retrieved + prescribed, land + water) retrievals into two (prescribed and retrieved surface), with the algorithm using the new land/water mask to determine the proper surface type.

Like most operational aerosol retrieval algorithms, this version of the MISR RA uses a pre-built lookup table (LUT) of radiative transfer (RT) output in lieu of running RT code on-the-fly. Previous versions of the MISR RA relied on either modified linear-mixing [*Abdou et al.*, 1997] or external-mixing of the phase functions [e.g., Limbacher and Kahn, 2019] to create aerosol mixture analogs from component particle optical analogs represented in our LUT. Although both approaches



tend to yield more accurate modeled TOA reflectances at higher AOD, external mixing requires the generation of massive LUTs containing thousands of mixtures to fully account for the range of aerosol properties found in nature, and modified linear mixing requires a significant computational cost to generate reasonably accurate upwelling radiances. To improve our sensitivity to aerosol type, we have built a new LUT of aerosol model components (Table 1) that when linearly mixed with

each other should more accurately account for the variability of aerosols seen in nature. This new component LUT contains TOA modeled reflectance data as a function of spectral band, solar/viewing geometry, AOD, aerosol optical model (or component), as well as surface pressure (for over-land retrievals) and 10m wind-speed (for over-water retrievals). Six-hourly wind-speeds are obtained from CCMP v2.0 data [*Mears et al*., 2019] and are spatially and temporally interpolated to the MISR domain and overpass time. The LUT values are interpolated during the retrieval process to the appropriate solar/viewing

geometry, surface pressure, and wind-speed.

Because the two sets of algorithms diverge from this point, section 2.1.1 describes the prescribed surface aerosol retrieval and section 2.1.2 delves into the retrieved surface aerosol retrieval.

### 2.1.1 MISR RA Prescribed Surface Aerosol Retrieval, *using Bounded Variable Least Squares (BVLS)*

As the name suggests, the MISR prescribed surface aerosol retrieval algorithm requires external data on both surface angular-

spectral reflectance and surface albedo for each individual MISR pixel. The process is summarized in supplemental Figure S1. Over-water, we assume that the surface reflectance is Lambertian (once glint is subtracted), with the prescribed surface albedos given in section 2.1. Because we do not use an over-water surface reflectance database (analogous to MAIAC over-land), our over-water prescribed surface results will likely be prone to error when aerosol loading is low. However, as described in 2.1.3 below, the combined surface algorithm addresses this limitation. Over land, the spectral albedo and angular dependence come

from MAIAC data that are bias corrected to remove artifacts that can originate in part from differences between the MISR and MODIS spectral band passes. A simple linear model was used for surface reflectance (and albedo) corrections in each MISR band, with the following slopes (m) and offsets (b) used for the blue, green, red, and NIR bands, respectively (m=[1.1, 1.1, 1.1, 1.0]; b=[0.015, 0.0, 0.0, 0.0]). These coefficients were identified by comparing retrieved surface albedos (section 2.1.2) with the prescribed albedos from MAIAC in regions where the MISR retrieved-surface-RA AOD agreed well with AERONET

AOD and AEROENT AOD <0.2. The fact that this bias correction was not sufficient to remove the AOD bias seen in the prescribed surface retrieval over-land (especially at AODs < 0.20) indicates that a camera-by-camera correction should probably be used in the future. However, because the primary focus of the prescribed surface aerosol retrieval is to improve our sensitivity to AOD and aerosol properties when aerosol loading is elevated (generally >1), we are not as concerned about the results of this retrieval when aerosol loading is low.

As our sensitivity to aerosol particle properties should be enhanced when optical loading is high specifically because we are prescribing the surface reflectance, the discrete set of mixtures used by the retrieved-surface algorithm (2.1.2) might be insufficient to describe the variability of aerosols seen in nature. Instead, we convert our component LUT (Table 1) into two regular grids composed of 15 fine-mode (FM) components and 2 coarse-mode components (as shown in Figure 1).



**Table 1: Microphysical and optical properties of new RA aerosol component climatology**

| Analog (aerosol type) | $r_e$ | $r_0$ | $r_1$ | ANG | SSA (550 nm) | AAE |
|---|---|---|---|---|---|---|
| Very small, spherical, strongly absorbing BlS | 0.06 | 0.001 | 0.4 | 2.74 | 0.80 | 1.43 |
| Very small, spherical, strongly absorbing BrS | 0.06 | 0.001 | 0.4 | 3.17 | 0.80 | 3.23 |
| Very small, spherical, moderately absorbing BlS | 0.06 | 0.001 | 0.4 | 2.97 | 0.90 | 1.35 |
| Very small, spherical, moderately absorbing BrS | 0.06 | 0.001 | 0.4 | 3.19 | 0.90 | 3.12 |
| Small, spherical, strongly absorbing BlS | 0.12 | 0.001 | 0.75 | 1.80 | 0.80 | 1.34 |
| Small, spherical, strongly absorbing BrS | 0.12 | 0.001 | 0.75 | 2.04 | 0.80 | 3.02 |
| Small, spherical, moderately absorbing BlS | 0.12 | 0.001 | 0.75 | 2.05 | 0.90 | 1.37 |
| Small, spherical, moderately absorbing BrS | 0.12 | 0.001 | 0.75 | 2.18 | 0.90 | 3.14 |
| Medium, spherical, strongly absorbing BlS | 0.26 | 0.01 | 1.5 | 0.69 | 0.80 | 0.91 |
| Medium, spherical, strongly absorbing BrS | 0.26 | 0.01 | 1.5 | 0.76 | 0.80 | 2.36 |
| Medium, spherical, moderately absorbing BlS | 0.26 | 0.01 | 1.5 | 0.92 | 0.90 | 1.08 |
| Medium, spherical, moderately absorbing BrS | 0.26 | 0.01 | 1.5 | 0.98 | 0.90 | 2.74 |
| Very small, spherical, non-absorbing | 0.06 | 0.001 | 0.4 | 3.22 | 1.00 | N/A |
| Small, spherical, non-absorbing | 0.12 | 0.001 | 0.75 | 2.31 | 1.00 | N/A |
| Medium, spherical, non-absorbing | 0.26 | 0.01 | 1.5 | 1.22 | 1.00 | N/A |
| Large, spherical, non-absorbing | 1.28 | 0.1 | 10 | -0.20 | 1.00 | N/A |
| Large, non-spherical, weakly absorbing | 1.48 | N/A | N/A | -0.03 | 0.96 | 2.71 |

**Column 1 describes the aerosol analogs, columns 2-4 represent effective radius, minimum radius, and maximum radius (respectively). Column 5 is Ångström exponent, column 6 in 550 nm single-scattering albedo (SSA), and the last column is absorption Ångström exponent (AAE). Spherical aerosol component optical properties are modeled using a Mie code (ref.), with an assumed log-normal particle size distribution. The non-spherical component optical model is described in (*Lee et al.* [2017]).**

Because our coarse-mode component list contains only one large spherical model and one large non-spherical model, the only coarse-mode parameter we attempt to retrieve is non-sphericity. This is done because (a) given the longest MISR wavelength is 867 nm, there is limited sensitivity to coarse-mode particle microphysical properties, and (b) we do not have very good optical models for dust and volcanic ash that often dominate coarse-mode particle types. We include five fine-mode particles in each of three size distributions, with 550 nm SSA values of 0.8 0.9, and 1.0, as well as flat (black smoke or BlS analog) and steep (brown smoke or BrS analog) SSA spectral dependence. We convert our 15 fine-mode component list into a three-dimensional grid of fine-mode effective radius ($r_e$), fine-mode 550 nm single-scattering albedo (SSA), and fine-mode BrS fraction (roughly analogous to fine-mode absorption angstrom exponent [AAE], though not identical). All told, the algorithm retrieves 550 nm AOD and the following five pieces of information related to aerosol microphysical/optical properties: 550 nm fine-mode fraction (FMF), 550 nm coarse-mode non-sphericity, fine-mode size ($r_e$; μm), 550 nm fine-mode SSA, and 550 nm fine-mode BrS fraction.

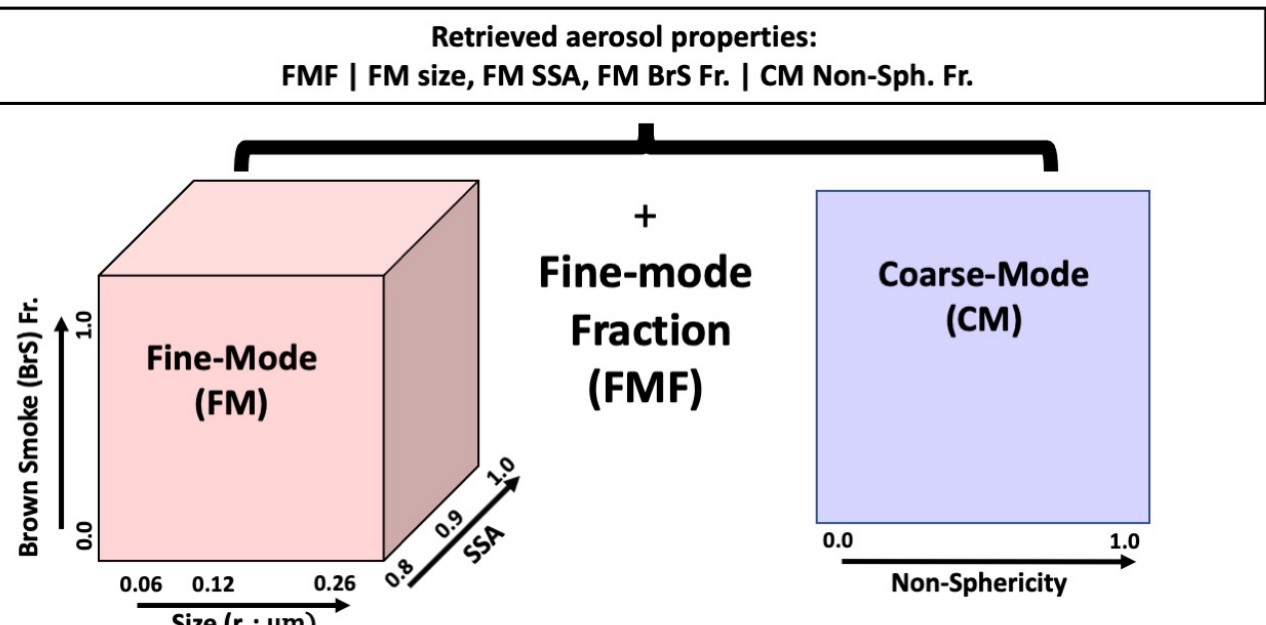

**Figure 1)** The left cube shows how our 15 fine-mode components are organized into a rectangular grid which yields three independent pieces of information (fine-mode effective radius, 550 nm SSA, and 550 nm BrC fraction. Because we assume that SSA is constant vs. wavelength when 550 nm SSA=1.0, we use the same models for both BrC=1 and BrC=0 for a SSA of unity (which yields 15 instead of 18 fine-mode models). The right square shows that our two coarse-mode components are broken down into a single coarse-mode non-sphericity parameter. We also retrieve fine-mode fraction, which represents the relative abundance of fine-mode aerosol loading to total aerosol loading at 550 nm.

Once we have converted our component LUT into two regular grids (fine and coarse modes), the algorithm then needs a starting point to begin iterating towards a solution. This initial guess is set to the following: AOD=0.10, FMF=0.8, coarse-mode non-sphericity=0.8, fine-mode size=0.1201 microns, fine-mode SSA=0.975, and fine-mode BrS fraction=0. The algorithm then interpolates the fine-and-coarse LUTs separately before linearly combining the modeled fine-and-coarse modes. For a given solution vector (AOD + aerosol properties), we generate 36 TOA modeled reflectances ($\rho_{\lambda,c}^{\mathrm{mod}}$), defined as:

$$\rho_{\lambda,c}^{\mathrm{mod}} = P_{\lambda,c} + \frac{ET_{\lambda,c} * Surf_{\lambda,c}}{1 - s_\lambda * A_\lambda}. \tag{2}$$

Here, $P_{\lambda,c}$ represents the modeled, interpolated path radiance, which is radiation that does not interact with the surface. To simplify the over-water algorithm, we also embed Fresnel reflection and whitecaps into this term. We estimate the TOA surface-reflected radiation as the normalized bottom-of-atmosphere downward irradiance multiplied by the azimuthally averaged surface-to-camera transmittance ($ET_{\lambda,c}$) multiplied by the surface reflectance ($Surf_{\lambda,c}$). We assume that the multiply reflected radiation can be accounted with the normalization (1-$s_\lambda$ *$A_\lambda$), where $s_\lambda$ represents the effective atmospheric backscatter and $A_\lambda$ represents the surface albedo.


We then calculate the derivatives of (2) with respect to all six aerosol-related parameters and set up our linear system of equations  The weighted linear system of equations ($\sqrt{w} \cdot A \cdot \vec{x} = \sqrt{w} \cdot \vec{b}$) can be described as:

$$
\begin{bmatrix} \sqrt{\frac{w_{1,1}}{Unc_{1,1}^2}} & \cdots & 0 \\ \vdots & \ddots & \vdots \\ 0 & \cdots & \sqrt{\frac{w_{4,9}}{Unc_{4,9}^2}} \end{bmatrix} \cdot \begin{bmatrix} \frac{\partial\rho_{1,1}^{mod}}{\partial Par_1} & \cdots & \frac{\partial\rho_{1,1}^{mod}}{\partial Par_6} \\ \vdots & \ddots & \vdots \\ \frac{\partial\rho_{4,9}^{mod}}{\partial Par_1} & \cdots & \frac{\partial\rho_{4,9}^{mod}}{\partial Par_6} \end{bmatrix} \cdot \begin{bmatrix} \Delta Par_1 \\ \vdots \\ \Delta Par_6 \end{bmatrix} = \begin{bmatrix} \sqrt{\frac{w_{1,1}}{Unc_{1,1}^2}} & \cdots & 0 \\ \vdots & \ddots & \vdots \\ 0 & \cdots & \sqrt{\frac{w_{4,9}}{Unc_{4,9}^2}} \end{bmatrix} \cdot \begin{bmatrix} (\rho_{1,1}^{TOA} - \rho_{1,1}^{mod}) \\ \vdots \\ (\rho_{4,9}^{TOA} - \rho_{4,9}^{mod}) \end{bmatrix}, \quad (3)
$$

where $\Delta Par_1$ represents the change in our retrieved first parameter (AOD; from its last guess), and $\Delta Par_6$ represents the change in our retrieved 6th parameter (Fine-mode Brown smoke fraction) compared to its initial guess or the result of the previous iteration. The derivative matrix (e.g., $\frac{\partial\rho_{1,1}^{mod}}{\partial Par_i}$) represents the change in modeled TOA reflectance with respect to a change in one of our retrieved parameters (such as AOD). The difference vector (column vector on the right) represents the difference between the observations and the current modeled TOA reflectances. On average, the magnitude of this vector

should decrease with every iteration as the algorithm converges to a better solution vector. The diagonal weight matrix (first matrix on the left on both sides of equation), which convolves channel weights (w) with their respective channel uncertainties (Unc), is used to account for things such as excessive sun-glint, topographic shadowing, and missing data, as well as accounting for the uncertainty in the model/measurement system (more detail on this can be found in Limbacher and Kahn, 2019). The fact that this is a diagonal matrix means that we assume our channel weights and uncertainties are uncorrelated (by channel).

Solving for the change in our retrieved parameter vector ($\overrightarrow{\Delta Par}$) is done using a bounded-variable least-squares (BVLS) solver (Lawson and Hanson, 1995), which allows us to put constraints on $\overrightarrow{\Delta Par}$ to ensure that our retrieved parameters stay within physical bounds (i.e., 0<AOD<10, 0<FMF<1.0, etc.). The iterative process of interpolating to a new model reflectance (2), calculating its derivatives, and then iterating to a more optimal solution (3) continues for a minimum of 5 iterations, until the change in our cost function,

$$
Cost = \frac{\sum_\lambda \sum_c \left( \frac{\sqrt{w_{\lambda,c}} * [\rho_{\lambda,c}^{TOA} - \rho_{\lambda,c}^{mod}]}{Unc_{\lambda,c}} \right)^2}{\sum_\lambda \sum_c w_{\lambda,c}}, \quad (4)
$$

falls below a certain tolerance (currently set to 0.00001), or 100 iterations have occurred. One of the problems with linear least-squares retrievals is that the assumed linearity in model response may not be accurate very far from where the derivatives

were calculated. This can result in the solution vector "bouncing around," slow convergence, or non-convergence. To address



this, if the algorithm detects that the cost function has not decreased after a new iteration, it multiplies the change in our retrieved parameter vector $(\overrightarrow{\Delta Par})$ by 0.5 and recomputes the cost function. The algorithm will continue doing this until the new cost function is lower than the value calculated for $\overline{\Delta Par} = 0$ (i.e., the cost function of the previous iteration).

Once the algorithm has converged to a solution, it converts the fine-and-coarse particle property grids back into a 1-dimensional list of 550 nm aerosol mixture fraction (for all 17 components), while also reporting 550 nm AOD, the prescribed surface albedo, and cost. This can be done because our list of 17 component aerosol particle analogs exactly maps to the bins shown in Figure 1. To decrease file size (which is still ~ 20 GB for all AERONET data in the validation dataset), we don't save the mixture fractions for all 17 components, but rather save information such as 550 nm fine-mode fraction, 550 nm SSA, etc. based on the aggregated results.

### 2.1.2 MISR RA Retrieved Surface Aerosol Retrieval, *using Discrete Aerosol Mixtures*

Although MODIS MAIAC-retrieved surface reflectance allows the MISR RA to retrieve AOD and aerosol properties over-land when aerosol loading is elevated, the quality of MISR RA retrievals is negatively impacted when the MAIAC surface is assumed, and aerosol loading is low-to-moderate (AOD at 550 nm <1). This is due factors such as differences between the MISR and MODIS spectral responses, gridding error, and MAIAC retrieved surface reflectance error (which should be much larger for the MISR 70˚-viewing cameras than for the near-nadir cameras). As a result, a version of the MISR RA was developed that self-consistently retrieves AOD, aerosol properties, and surface properties at pixel-level resolution (1 km here). The MISR RA retrieval surface algorithm is functionally identical to the algorithm described in Limbacher and Kahn [2019] with the following two exceptions, described briefly below: 1) a modification of the discrete list of aerosol mixtures used by the retrieval algorithm, and 2) the addition of an over-land retrieval.

As in *Limbacher and Kahn* [2019], we use the same exponential weighted average of discrete aerosol mixtures (at their best fitting AOD) to identify aggregate aerosol and surface properties. However, the discrete aerosol mixtures we use for this technique have been updated to reflect our new component climatology. As in section 2.1.1, we break up our components into 15 fine-mode components and 2 coarse-mode components. Here, we also remove all brown-smoke component analogs, as we are unlikely to have sensitivity to brown vs. black smoke fraction for the low-moderate AOD regime where this algorithm will be most useful. The nine-remaining fine-mode components are then permitted to mix with the 2 coarse-mode components in increments of 20%, resulting in a total of 65 discrete aerosol mixtures with TOA modeled reflectances that can still be appropriately described by equation 2. A flow chart describing this new retrieval is presented in supplemental as Figure S2; we provide a short summary of the technique below.

The addition of an over-land retrieval to the MISR RA retrieved-surface algorithm represents a relatively simple extension and upgrade of our existing over-water retrieval that allows for shallow, turbid, and eutrophic water, as described in *Limbacher and Kahn* [2019]. For both the over-land and over-water MISR RA retrieved-surface algorithms, we first redefine the surface reflectance as follows:



$$\text{Surf}_{\lambda,c} = A^*_\lambda * \text{L}_c; \quad A^*_\lambda = \frac{A_\lambda}{1-s_\lambda*A_\lambda}, \tag{5}$$

where $A_\lambda$ represents the view-invariant surface albedo and $L_c$ represents the spectrally invariant angular brightness coefficient (this is set to 1.0 for over-water retrievals). $A^*_\lambda$ represents a reasonably accurate estimate of the impacts of including multiple reflections into our modified surface albedo, as this significantly simplifies the surface retrieval with no adverse impacts (we

disentangle this term later). Equation 5 is also known as a shape-similarity assumption because the spectral surface reflectance is assumed to vary by the same relative fraction at each view-angle (surface brightness can change with view angle, but its color cannot). This shape-similarity assumption has its heritage in the multiangle Along-Track Scanning Radiometer-2 (ATSR-2) instrument (Flowerdew and Haigh [1995]; Veefkind et al. [1998]) and was adopted by the MISR team as part of the MISR standard aerosol retrieval algorithm (Diner et al., 2005).

To retrieve the surface reflectance for any given AOD and aerosol model, we rewrite our cost function using Equations 2 and 4 by applying the shape-similarity assumption (Equation 5):

$$\text{Cost} = \frac{\sum_\lambda \sum_c \left( \frac{\sqrt{w_{\lambda,c}} * \left[ \rho^{\text{TOA}}_{\lambda,c} - \left( P_{\lambda,c} + \text{ET}_{\lambda,c} * A^*_\lambda * \text{L}_c \right) \right]}{\text{Unc}_{\lambda,c}} \right)^2}{\sum_\lambda \sum_c w_{\lambda,c}}. \tag{6}$$

For every AOD and aerosol model in our LUT, we first get an estimate of the modified surface albedo ($A^*_\lambda$) by assuming that the surface can be adequately described as Lambertian, which requires that we set $L_c=1$. We then take the derivative of (6)

with respect to $A^*_\lambda$ (here, we assume $\frac{\partial \text{L}_c}{\partial A^*_\lambda} = 0$), set the result to 0, and analytically solve for the modified surface albedo,

$$A^*_\lambda = \frac{\sum_c \frac{w_{\lambda,c}}{\text{Unc}^2_{\lambda,c}} * \text{ET}_{\lambda,c} * \text{L}_c * \left[ \rho^{\text{TOA}}_{\lambda,c} - P_{\lambda,c} \right]}{\sum_c \frac{w_{\lambda,c}}{\text{Unc}^2_{\lambda,c}} * \left[ \text{L}_c * \text{ET}_{\lambda,c} \right]^2}. \tag{7}$$

For our over-water retrieval, this is the only step required to estimate the modified surface albedo for a given AOD and aerosol mixture. However, over land, we must solve for the shape-similarity coefficient ($L_c$) by taking the derivative of (6) with respect to $L_c$, setting it equal to 0 (here we assume $\frac{\partial A^*_\lambda}{\partial \text{L}_c} = 0$), and solving for $L_c$:

$$\text{L}_c = \frac{\sum_\lambda \frac{w_{\lambda,c}}{\text{Unc}^2_{\lambda,c}} * A^*_\lambda * \text{ET}_{\lambda,c} * \left[ \rho^{\text{TOA}}_{\lambda,c} - P_{\lambda,c} \right]}{\sum_\lambda \frac{w_{\lambda,c}}{\text{Unc}^2_{\lambda,c}} * \left[ A^*_\lambda * \text{ET}_{\lambda,c} \right]^2}. \tag{8}$$

For our over-land retrieval, we then iterate through equations (7) and (8) twice, as the algorithm typically converged after two iterations (based on prior experience), which results in further refinement of both $A^*_\lambda$ and L.





Following Figure 3 and as summarized above, we retrieve the modified surface albedo ($A_\lambda^*$) and shape-similarity coefficient ($L_c$) for all 65 discrete aerosol mixtures and 26 AODs found in our RT LUT (Table 2). To iterate towards the optimum AOD for each of those 65 aerosol mixtures, the algorithm also temporarily saves information such as cost function (65 mixtures x 26 AODs) and channel-specific residual (65 mixtures x 26 AODs x 4 bands x 9 cameras). These channel-
specific residuals are simply the portion of our cost-function (equation 6) found within the outer parentheses ($\frac{\sqrt{w_{\lambda,c}}*[\rho_{\lambda,c}^{TOA}-(P_{\lambda,c}+ET_{\lambda,c}*A_\lambda^**L_c)]}{Unc_{\lambda,c}}$). After computing this information on the coarse grid found in our LUT, the algorithm then iterates towards a better-fitting (and more precise) AOD and surface for each of the 65 aerosol mixtures using a bisectional approach with 5 iterations; given the coarse-grid spacing shown in Table 2, the resulting AOD should have an algorithmic precision ranging from <0.001 at an AOD of 0.0 to ~0.025 at and AOD of 10.

Once the optimum AOD and surface reflectance properties have been calculated for each aerosol mixture, normalized mixture weights are calculated according to

$$MW_m = \frac{\exp\left(\frac{Cost_{min}-Cost_m}{Cost_{min}+0.01}\right)}{\Sigma_m\left[\exp\left(\frac{Cost_{min}-Cost_m}{Cost_{min}+0.01}\right)\right]}, \tag{9}$$

where the subscript $m$ represents aerosol mixture, $Cost_m$ represents the lowest cost (best fit) for each of the 65 aerosol mixtures, and $Cost_{min}$ represents the lowest cost among all mixtures. Weighted aggregate parameters are then calculated for the following: 550 nm AOD, modified surface albedo ($A_\lambda^*$), shape-similarity coefficient ($L_c$), aerosol component fraction (Table
1), and cost. Finally, $A_\lambda^*$ is corrected for multiple reflections via division by $(1.0 + s_\lambda * A_\lambda)$. As in the previous section, the algorithm then converts aerosol component fraction into fine-mode fraction, ANG, and SSA while also reporting 550 nm AOD, the retrieved surface albedo, and cost.

Over water, this algorithm retrieves 5 pieces of information about aerosol loading/properties and 4 pieces of information about the surface spectral reflectance ($A_\lambda$). Over land, the algorithm retrieves an additional 9 pieces of information
about the surface reflectance angular behavior, which yields a total of 18 retrieved pieces of information from 36 measurements. Even in the most topographically complex regions (where up to four MISR cameras may be eliminated due to obscuration) the number of observations will exceed the number of retrieved parameters. A major limiting factor of this algorithm is the assumption of surface shape-similarity. If the color of the surface changes with view-angle, as it does in some desert regions, the algorithm will alias those errors into the retrieved aerosol properties and AOD.

**2.1.3 MISR RA Combined Surface Aerosol Retrieval**

The prescribed and retrieved surface approaches are as described in sections 2.1.1 and 2.1.2. Over land, the combined surface approach uses the prescribed surface AOD to identify the optimal retrieval type for a given pixel. If the prescribed surface AOD is less than 1.0, the combined surface retrieval selects the AOD and aerosol properties from the retrieved surface algorithm. If the prescribed surface AOD is greater than 2, the combined surface retrieval selects the AOD and





aerosol properties from the prescribed surface algorithm. If the prescribed surface AOD falls between 1 and 2, the combined surface retrieval linearly interpolates AOD and aerosol properties between the two algorithms. The logic behind this combined surface retrieval is two-fold. When aerosol loading is low, errors in the surface reflectance based on the prescribed surface retrieval tend to produce significant high biases in AOD and errors in aerosol particle properties. Conversely, when

aerosol loading is high, the retrieved surface algorithm is unable to properly separate the surface and atmospheric contributions, leading to a substantial low bias in AOD [*Kahn et al.*, 2010]. Empirically, we find this approach with these domain boundaries also yields optimal results when compared to AERONET, as shown in Section 3 below.

Over water, the combined surface algorithm is used, with the same AOD constraints as described above. However, because our prescribed surface could be very inaccurate (and result in low-quality aerosol retrievals), the algorithm instead

uses the retrieved surface AOD (instead of the prescribed surface AOD, as is done over land) to determine the algorithm type to be used for the final aerosol result (prescribed surface, retrieved surface, or combined surface). Even though the retrieved surface algorithm suffers from an AOD low bias at high AOD, the retrieved surface algorithm still appears to retain sensitivity to AOD even when AERONET AOD exceeds 3, which makes this algorithm suitable for determining the algorithm type used. Due to the low numbers of high AOD MISR/AERONET coincidences over water, the combined

surface AOD bounds (1 and 2) may need to be modified when we have more data (or if we begin using a surface reflectance dataset for our prescribed-surface over-water retrievals).

## 2.2 MISR RA Updated Aerosol Component Climatology

The updated LUT containing RT output was created using SCIATRAN version 3.8 (*Rozanov et al.* [2014], https://www.iup.uni-bremen.de/sciatran/index.html, last accessed 8/17/2020). The RT code was run using the full-vector

discrete ordinates method solver with 16 streams for our fine-mode optical analogs and 32 streams for our two coarse-mode analogs. Detailed information about our 17 updated aerosol components can be found in Table 1 and information about the size and dimensionality of the LUT are given in Table 2. Even though Table 2 appears to have eight dimensions, the LUTs are broken up into a 7-dimensional over-water LUT (pressure is assumed to be 1013.25 mb) and a 7-dimensional over-land LUT (no wind-speed dimension needed). The goal in creating the individual aerosol components shown in Tables 1 and 2 is

to capture aerosol particle property variability in as few components possible, under the assumption that we can linearly mix the radiances of these mixtures to create a continuum in terms of aerosol size, shape, and single-scattering albedo. For our spherical absorbing analogs, we now include aerosol sizes ranging from 0.06 to 0.26 microns effective radius, which adds analogs that were missing in our 2014 dataset (*Limbacher and Kahn* [2014]) and from the operational MISR product (*Kahn et al.* [2010]). Previously, we used a dust model optimized for the red and NIR channels only (*Kalashnikova et al.* [2005]),

so we replaced it with one that is modeled consistently for all MISR spectral bands (*Lee et al.*, [2017]), as described in section 2.2.1 below.



**Table 2: Updated LUT values and dimensionality.**

| Component name (17) | 550 nm AOD (26) | λ (nm) (7) | $\mu_0$ (10) | μ (8) | Δϕ (19) | 10-m wind (m/s) (5) | Surface pressure (mb) (2) |
|---|---|---|---|---|---|---|---|
| sph_abs_0.06_0.80_BlS | 0 | 446.34 | 0.1 | 0.3 | 0 | 1 | 608 |
| sph_abs_0.06_0.80_BrS | 0.05 | 557.54 | 0.2 | 0.4 | 10 | 5 | 1050 |
| sph_abs_0.06_0.90_BS | 0.1 | 671.75 | 0.3 | 0.5 | 20 | 8 | |
| sph_abs_0.06_0.90_BrS | 0.15 | 866.51 | 0.4 | 0.6 | 30 | 12 | |
| sph_abs_0.12_0.80_BlS | 0.25 | | 0.5 | 0.7 | 40 | 20 | |
| sph_abs_0.12_0.80_BrS | 0.35 | | 0.6 | 0.8 | 50 | | |
| sph_abs_0.12_0.90_BlS | 0.5 | | 0.7 | 0.9 | 60 | | |
| sph_abs_0.12_0.90_BrS | 0.65 | | 0.8 | 1 | 70 | | |
| sph_abs_0.26_0.80_BlS | 0.85 | | 0.9 | | 80 | | |
| sph_abs_0.26_0.80_BrS | 1.05 | | 1 | | 90 | | |
| sph_abs_0.26_0.90_BlS | 1.3 | | | | 100 | | |
| sph_abs_0.26_0.90_BrS | 1.55 | | | | 110 | | |
| sph_nonabs_0.06 | 1.85 | | | | 120 | | |
| sph_nonabs_0.12 | 2.15 | | | | 130 | | |
| sph_nonabs_0.26 | 2.5 | | | | 140 | | |
| sph_nonabs_1.28 | 2.85 | | | | 150 | | |
| Dust | 3.25 | | | | 160 | | |
| | 3.65 | | | | 170 | | |
| | 4.1 | | | | 180 | | |
| | 4.55 | | | | | | |
| | 5 | | | | | | |
| | 5.65 | | | | | | |
| | 6.45 | | | | | | |
| | 7.35 | | | | | | |
| | 8.5 | | | | | | |
| | 10 | | | | | | |

**Each column lists the values of the variable in the heading that are included in the LUT. The number of values is given in parentheses at the top, The overall dimensionality of the LUT is eight, although it is broken up into a 7-dimensional over-land LUT (no wind-speed dimension; 9.4 x $10^6$ elements) and a 7-dimensional over-water LUT (surface pressure assumed to be 1013.25 mb; 2.35 x $10^7$ elements).**


### 2.2.1 Updated Dust Optical Model

The non-spherical dust optical model used in the RA is created following *Lee et al.* [2017], except with the MISR spectral bands. The non-spherical dust's phase matrix (for all spectral bands) is derived by integrating the single-scattering properties of individual non-spherical particles over both size and shape distributions. Thus, representative size/shape distributions and the spectral refractive indices for dust are determined from Aerosol Robotic Network (AERONET; *Holben et al.*, [1998]) inversion data at Capo Verde for heavy dust events (coarse-mode AOD > 0.5 and FMF < 0.2), with the medians of the data record taken as representative values. Note that the AERONET inversion assumes a fixed spheroid shape mixture (*Dubovik et al.*, [2006]), and thus the same is used for consistency. The single-scattering properties of individual spheroids are available from an aerosol single-scattering property database (*Meng et al.*, [2010]) enabling one to easily obtain the spectral optical properties of dust. Similar dust models have been widely used in various aerosol retrieval algorithms, as they improve artificial biases in AOD and Ångström exponent (ANG) retrievals due to inaccurate representation of non-spherical dust by spherical aerosol modeled with Mie theory (*Dubovik et al.*, [2014]; *Hsu et al.*, [2019]; *Lee et al.*, [2012, 2017]; *Lyapustin et al.*, [2018]; *Sayer et al.*, [2018]; *Zhou et al.*, [2020]).

### 2.3 AERONET data and validation methodology

With hundreds of sites scattered worldwide, AERONET sun photometers directly measure spectral AOD (*Holben et al.*, 1998) at an uncertainty of ~0.01 (*Eck et al.*, 1999; *Sinyuk et al.*, 2012), and offer excellent cloud-screening as part of the version 3 algorithm (*Giles et al.*, 2019). Provided that AOD is >~ 0.1 or 0.2), AERONET ANG can also be reported very accurately (*Wagner and Silva*, 2008). As in *Limbacher and Kahn* (2019), we first interpolate AERONET AOD (here we use L1.5 AOD, as cloud screening for L1.5 is much better in version 3 that previous versions, and offers many more retrieval results than the L2 products) to the MISR band centers, using a second-order polynomial in log-space. We then compute Ångström exponent as a log-log fit of interpolated AOD to wavelength, using all four MISR wavelengths. For the AERONET direct-sun parameters (AOD and ANG), we attempt to limit spatio-temporal variability from negatively impacting our comparison with MISR by masking out all AERONET data falling outside a ±30-minute window centered on the MISR overpass. AERONET AOD and ANG at 550 nm are then averaged over this window prior to comparison with the MISR RA.

Although AERONET almucantar inversions (*Dubovik and King*, 2000) represent retrievals of aerosol properties such as coarse-mode sphericity and SSA rather than direct measurements, they provide an opportunity to compare with aerosol particle properties retrieved from imagers such as MISR over diverse regions and temporal ranges that can span more than a decade. Because almucantar inversions are performed far less frequently than AOD is sampled, we limit potential coincidences to within ±4 hours of the MISR overpass time, saving the following averaged (mean) 550 nm parameters: absorbing AOD, fine-mode AOD, coarse-mode AOD, and sphericity. Average single-scattering albedo is then calculated as absorbing AOD/(fine-mode AOD + coarse-mode AOD). Fine-mode fraction (FMF) is calculated as fine-mode AOD/(fine-





mode AOD + coarse-mode AOD). Additionally, because the MISR RA retrieves sphericity only for the coarse mode, we consider only the coarse-mode component of AERONET non-sphericity, calculated as (1.0-FMF)*(1.0-sphericity/100).

## 3 Results

### 3.1 MISR RA Over-Land Validation using AERONET

5 As explained in the previous section, we use a ±30-minute averaging window for comparing AERONET direct-sun results with the MISR RA and a ±4 hour averaging window for comparing AERONET almucantar inversion results with the MISR RA. Because retrieval quality likely degrades dramatically in the presence of clouds, sea-ice, bright desert, and where retrieval fits are poor (i.e., a high cost function), we established a series of tests to help identify good-quality retrievals (for all 48x48 MISR RA retrievals centered on an AERONET station). Quality flags are set for each test.

1. MISR surface height (from the SA digital elevation map) is within 200m of the given AERONET station height
2. At least 7 of 9 MISR cameras contain valid radiance data
3. MISR pixel must be masked as land
4. MISR combined surface cost function < 1

5. MISR combined surface AOD < 9
6. $2^{nd}$ derivative of prescribed surface cost function with respect to AOD > 10
7. Normalized difference vegetation index (NDVI) using prescribed surface albedos > 0.1
8. Blue reflectance max – blue reflectance min (over all cameras) < 0.1 + 0.2*exp(-1.0*[MISR prescribed surface AOD])

9. MISR retrieved surface AOD standard deviation among all QA pixels < 1

Quality flag 1 just makes sure that we compare pixels at roughly the same elevation to each other (as dust and other aerosols tend to be concentrated in layers) and is only used when comparing AERONET AOD to MISR retrieved AOD. The reasoning here is that the total column loading will likely be different at difference surface elevations, but aerosol particle
25 properties will not vary as much. Quality flag 2 makes sure that a retrieval has enough "good" input data to give high-quality output, and quality flag 3 uses our previously computed land/water mask as we are only comparing the land algorithm to AERONET for the current validation exercise. Quality flag 4 uses the combined retrieval goodness-of-fit (cost) to screen out poor-quality (mostly cloud-contaminated) retrievals. Quality flag 5 indicates that results with a combined retrieval AOD greater than 9 are likely cloud. As we saw in *Limbacher and Kahn* [2019], the $2^{nd}$ derivative of our cost function can be a
30 good indicator of retrieval quality. A larger $2^{nd}$ derivative corresponds to a steeper minimum in our cost function with respect to AOD; we use 10 as a lower bound here in quality flag 6 as this tends to mask out some lower quality results (mostly clouds). Quality flag 7 primarily masks desert, unmasked water, and clouds using the MAIAC prescribed surface albedos





(these are input into the prescribed surface retrieval). Here, NDVI is calculated as the following: NDVI=(NIR-Red)/(NIR+Red). Quality flag 8 is used to mask partially cloudy MISR data (clouds in some cameras but not others), as the difference between the maximum and minimum reflectance will be quite large for such pixels. Quality flag 9 attempts to remove stray clouds via a large-scale (low frequency) variability filter.

**3.1.1 AERONET Direct Sun Validation of MISR Over-Land RA**

Applying the flags described in 3.1 and requiring at least 10 quality-assessed retrievals (out of 2304 potential, from 48x48 pixel patches) for each MISR/AERONET coincidence results in 9680 averaged MISR RA/AERONET over-land coincidences for the 4 years of processed MISR data, interspersed between September 2000 and November 2016. AOD statistics for the MISR/AERONET validation are shown in Table 3, and are provided for the retrieved-surface, prescribed

surface, and a combined surface approach.

**Table 3: MISR RA vs. AERONET direct-sun statistics over-land**

| AOD Comparison | # | RMSE | MAE | bias | r |
|---|---|---|---|---|---|
| Retrieved Surface (RS) | 9680 | 0.125 | 0.035 | -0.017 | 0.867 |
| Prescribed Surface (PS) | 9680 | 0.192 | 0.104 | 0.137 | 0.891 |
| Combined Surface (CS) | 9680 | 0.091 | 0.035 | -0.006 | 0.933 |
| | | | | | |
| ANG Comparison (CS Only) | # | RMSE | MAE | bias | r |
| CS ANG ∣ CS AOD>0.05 | 7112 | 0.422 | 0.284 | 0.072 | 0.501 |
| CS ANG ∣ CS AOD>0.20 | 2553 | 0.400 | 0.279 | 0.166 | 0.717 |
| CS ANG ∣ CS AOD>0.50 | 565 | 0.359 | 0.250 | 0.171 | 0.852 |
| CS ANG ∣ CS AOD>1.0 | 127 | 0.352 | 0.253 | -0.022 | 0.891 |

The rows under "AOD Comparison" indicate the type of MISR retrieval being compared to AERONET. The rows under "ANG Comparison" indicate the MISR RA AOD constraints being placed on the comparison with AERONET (MISR AOD must be > 0.05, etc). For the first column, RS corresponds to the MISR RA over-land Retrieved Surface algorithm, PS corresponds to the 15  MISR RA over-land Prescribed Surface algorithm, and CS corresponds to the MISR RA over-land Combined Surface algorithm. The number of MISR/AERONET coincidences used to generate a given set of statistics is given in column 2 (#). Root-mean squared error (RMSE) is given in column 3, median absolute error (MAE) is given in column 4, the average MISR-AERONET bias is given in column 5, and the Pearson correlation coefficient (r) is given in column 6.

Figure 2 shows the MISR/AERONET over-land 550 nm AOD comparisons for the retrieved surface (a, d),

prescribed surface (b, e), and the combined surface approach (c, f). Comparisons are plotted in both linear and log space, as it is easier to evaluate the lower AOD comparisons with a log-log plot. Additionally, the red lines on the log-log plots correspond to $\pm$ (0.225*[MISR AOD] + 0.025), which is our estimate of the expected error of the combined retrieval (Figure 3b). Figure 2d clearly demonstrates that the retrieved surface algorithm works quite well when AERONET AOD is low-moderate, whereas Figure 2b clearly shows the superiority of the prescribed surface algorithm when aerosol loading is high.

The combined approach described in section 2.1.3 leverages the strengths of each algorithm, resulting in an RMSE (0.091), 27% lower than the retrieved surface approach (0.125), 53% lower than the prescribed surface approach (0.192), and





yielding a correlation coefficient (r=0.933) that is higher than either approach. Because the statistics for the combined approach are significantly better than either the prescribed or retrieved surface approaches, the rest of the over-land validation shows only results from the combined surface approach.

Figure 3a presents a larger AOD scatterplot image of the MISR RA combined approach with the different algorithm

5    regimes color coded. Because the different regimes are based on the MISR prescribed surface AOD (not the combined AOD), the background color codes are approximate. Comparing to Figure 2, 3a clearly demonstrates that the combined approach is picking the best pieces from both algorithms. This algorithm also eliminates the tendency for the MISR RA (and the MISR SA) to underestimate AOD when aerosol loading is elevated. Figure 3b demonstrates that a **prognostic error** of ±0.225*[MISR AOD] ± 0.025 fits very well to the data, although this can be significantly reduced by applying further

10   quality constraints to the data. As a prognostic error, this can be used to estimate pixel-level uncertainty of MISR RA AOD without the use of AERONET data (assuming the data are cloud/quality screened in the manner described above and that AEROENT cloud-screening is not significantly biasing the results).



**Figure 2) Comparison of MISR RA over-land 550 nm AOD retrievals with AERONET direct-sun 550 nm AOD. x-axes represent AERONET 550 nm AOD and y-axes represent MISR RA retrieved 550 nm AOD. MISR over-land retrieval type (prescribed, retrieved, or combined) embedded in the lower right of each panel. Panels a) and d) present the MISR retrieved surface algorithm, b) and e) present the MISR prescribed surface algorithm, and c) and f) present the combined surface algorithm. Panels a-c) on the left show scatterplots of MISR RA AOD compared to AERONET, with a linear scale to allow for easier interpretation of high-AOD results. Panels d-f) on the right show 2-dimensional histograms of MISR RA AOD compared to AERONET, with a logarithmic scale to allow for easier interpretation of low-AOD results. Expected error of MISR combined surface AOD ±(0.225*AOD + 0.025) embedded as two red lines in panels d-f.**





**Figure 3)** Panel a) is identical to Figure 2g, with the addition of a color code added to identify regions where the prescribed and/or retrieved surface are used in the combined surface approach. Figure 3b shows MISR RA 68th percentile absolute AOD errors as a function of MISR combined over-land AOD. Data are plotted at increments of 2% (~160 coincidences per data point), with a black expected error line (derived from this data) plotted on top of the data.



Figure 4 compares the MISR combined retrieval ANG to AERONET ANG for MISR retrieved AOD greater than 0.20 (6a), 0.50 (6b), 1.0 (6c), and 1.5 (6d). Statistics for these plots are also provided in Table 3. Figure 4 (all panels) show two clusters of ANG for AERONET, one at ~0.25 (likely dust dominated), and another at ~1.5 (probably smoke/pollution dominated). The MISR RA captures the smoke/pollution dominated cluster very well but tends to over-estimate ANG

substantially as AERONET ANG decreases below ~1.25. Among several likely causes for this: 1) the MISR RA aerosol climatology currently contains no fine-mode non-absorbing analogs (or fine-mode dust analogs) with ANG lower than 1.22, 2) MISR's ~866 nm NIR band is too short to give an optimal spectral lever for large aerosols, 3) over land, the TOA signal at 866 nm tends to be dominated by the surface, resulting in minor surface reflectance errors aliasing into large errors in retrieved AOD and aerosol properties, and 4) errors in linear-mixing of fine-and-coarse mode modeled reflectances might

cause biases in retrieved ANG (and FMF). Regardless of the reasons for the discrepancies with AERONET, ANG RMSE of 0.407 and a correlation coefficient of 0.717 for MISR AOD >0.20 suggests that the algorithm still offers useful particle size constraints over land.

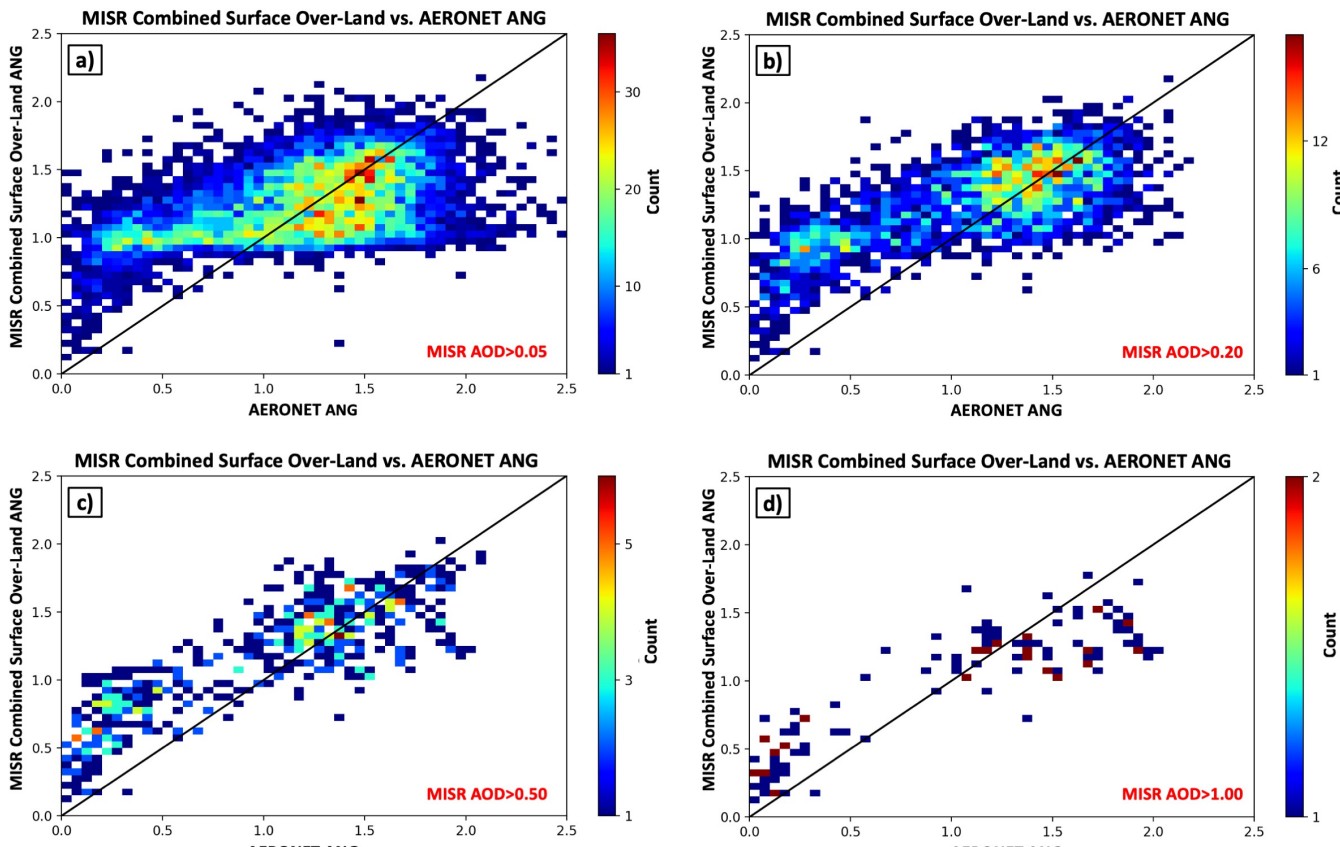

**Figure 4) 2-dimensional histograms of MISR RA combined-surface over-land ANG compared to AERONET ANG. x-axes are**
15 **AERONET ANG and y-axes are MISR combined-surface over-land ANG. Panel a) shows MISR ANG vs AERONET ANG, constrained by MISR retrieved AOD > 0.20. Panels b-d) show the same as a), but for MISR retrieved AOD constraints of >0.5 b), >1.0 c), and >1.5 d).**





### 3.1.2 AERONET Inversion Validation of MISR Over-Land RA

Using the retrieval quality flags indicated in section 3.1, combined with a 4-hour averaging window and 10-pixel minimum
(same as in section 3.1.1), yields 2332 MISR/AERONET inversion coincidences with MISR AOD > 0.2, 505 coincidences
with AOD > 0.5, and 107 coincidences with AOD > 1. Statistics for all figures shown (SSA, FMF, and coarse-mode non-
sphericity) can be found in table 4.

**Table 4: MISR RA vs. AERONET almucantar inversion statistics over-land**

| **550 nm FMF Comparison** | **#** | **RMSE** | **MAE** | **bias** | **r** |
|---|---|---|---|---|---|
| FMF \| 0.20>AOD>0.50 | 1827 | 0.194 | 0.132 | 0.022 | 0.597 |
| FMF \| 0.50>AOD>1.00 | 398 | 0.186 | 0.113 | 0.066 | 0.825 |
| FMF \| 1.00>AOD>1.50 | 62 | 0.107 | 0.060 | 0.047 | 0.968 |
| FMF \| AOD>1.50 | 45 | 0.087 | 0.033 | -0.010 | 0.971 |
| | | | | | |
| **Non-Sph. Fr. Comparison** | **#** | **RMSE** | **MAE** | **bias** | **r** |
| Non-Sph. Fr. \| 0.20>AOD>0.50 | 1827 | 0.238 | 0.103 | -0.087 | 0.653 |
| Non-Sph. Fr. \| 0.50>AOD>1.00 | 398 | 0.228 | 0.092 | -0.112 | 0.835 |
| Non-Sph. Fr. \| 1.00>AOD>1.50 | 62 | 0.149 | 0.056 | -0.057 | 0.936 |
| Non-Sph. Fr. \| AOD>1.50 | 45 | 0.070 | 0.022 | 0.007 | 0.983 |
| | | | | | |
| **550 nm SSA Comparison** | **#** | **RMSE** | **MAE** | **bias** | **r** |
| SSA \| 0.20>AOD>0.50 | 1827 | 0.048 | 0.032 | -0.007 | 0.330 |
| SSA \| 0.50>AOD>1.00 | 398 | 0.038 | 0.023 | 0.000 | 0.324 |
| SSA \| 1.00>AOD>1.50 | 62 | 0.026 | 0.017 | -0.006 | 0.430 |
| SSA \| AOD>1.50 | 45 | 0.019 | 0.013 | 0.000 | 0.807 |

**Same as Table 3, except for MISR RA vs AERONET inversion statistics over-land. All MISR data corresponds to the combined
surface retrieval. Note that AERONET inversion results are not ground truth, they represent retrieval results. The AERONET
team cautions against the use of results when blue-band AOD <0.4, so comparisons for green band AOD <0.50 should be**
**considered qualitative rather than quantitative.**

A comparison of MISR RA 550 nm FMF and AERONET 550 nm FMF is presented in Figures 5a (0.5<AOD<1.0),
5b (1.0<AOD<1.5), and 5c (AOD>1.5). Panels 5a-5c shows very similar patterns as compared to Figure 4, with excellent
sensitivity to retrievals of small (fine-mode) smoke and pollution aerosol, and less sensitivity seen in the coarse-mode-
dominated regions. However, as demonstrated in Figure 5b, 5c, and table 4, overall sensitivity to FMF increases substantially
for retrieved 1.0>AOD >1.5 (compared to 0.5>AOD>1.0), with RMSE dropping from 0.186 to 0.107, median absolute error
(MAE) improving from 0.113 to 0.06, and the correlation coefficient increasing from 0.825 to 0.968.





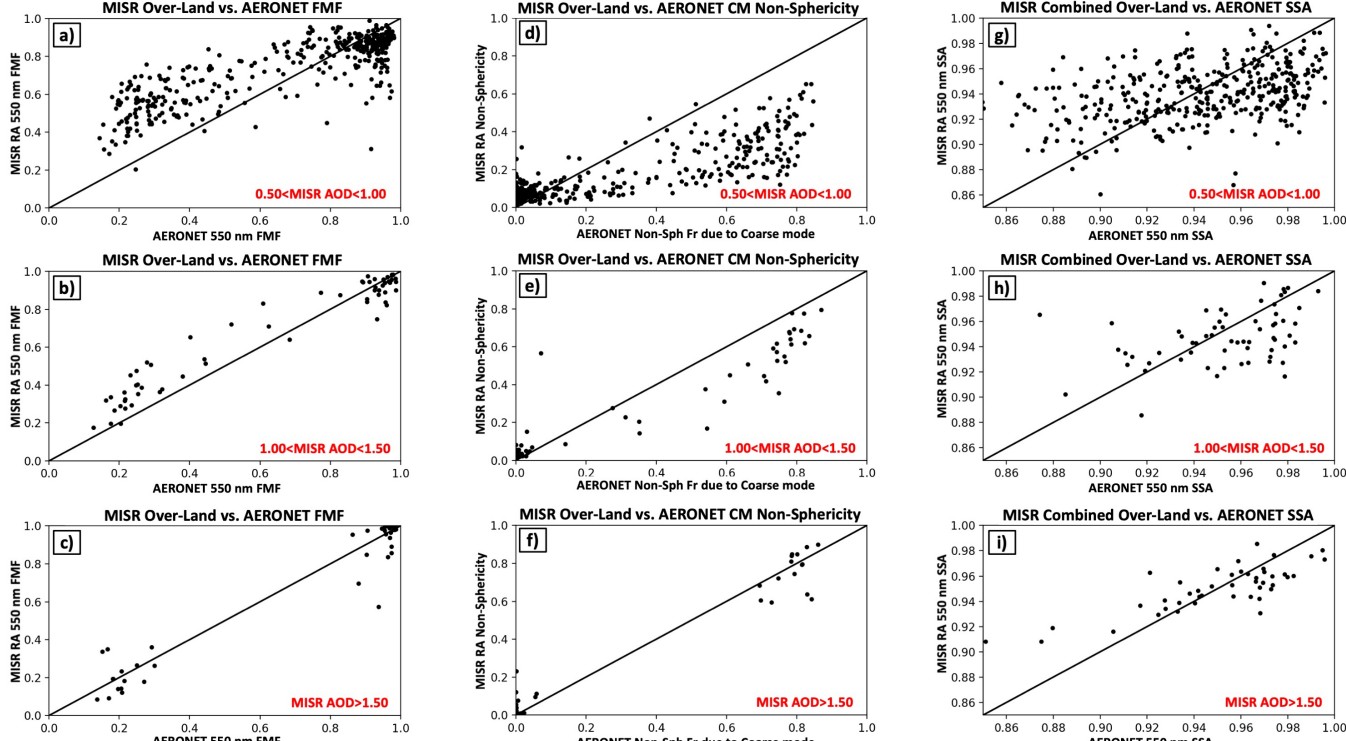

**Figure 5) Scatterplots of MISR RA combined-surface over-land 550 nm particle properties compared to AERONET 550 nm retrieved particle properties. x-axes are AERONET 550 nm particle properties and y-axes are MISR combined-surface over-land 550 nm particle properties. AOD constraints are embedded in red for each panel in the lower right corner. The first row of panels (a, d, g) corresponds to retrievals with 0.50<MISR AOD<1. The second row of panels (b, e, h) corresponds to retrievals with 1.00<MISR AOD<1.50. The third row of panels (c, f, i) corresponds to retrievals with MISR AOD>1.50. The first column of panels (a-c) show MISR RA retrieved fine-mode fraction (FMF) vs AERONET retrieved FMF, the second column of panels (d-f) show MISR RA retrieved non-spherical fraction vs AERONET retrieved non-spherical fraction attributed to the coarse mode (CM), and the third column of panels (g-i) show MISR RA retrieved single-scattering albedo (SSA) vs AERONET retrieved SSA.**

Because the current MISR RA has no fine-mode non-spherical component, the algorithm tends to dramatically underestimate retrieved non-spherical fraction compared to the value retrieved from AERONET. As described in section 2.3, we first convert the total column sphericity parameter retrieved by AERONET into non-spherical contribution due to the coarse mode via (1.0-FMF)*(1.0-sphericity/100). This allows for an apples-to-apples comparison of non-sphericity between the MISR RA and AERONET, and we refer to this parameter as non-sphericity or non-spherical fraction for the rest of the manuscript. Because this bounds the non-spherical fraction to between 0.0 and 1.0-FMF, the non-spherical comparison provided in Figure 5 (panels d-f) looks very similar to Figure 5 (panels a-c; 1.0 minus the values presented will make it look nearly identical). Just like FMF, MISR sensitivity to the non-spherical fraction over-land dramatically improves as AOD increases. Imposing more stringent AOD constraints (1.0>AOD >1.5 compared to 0.5>AOD>1.0), RMSE drops from 0.228 to 0.149, MAE drops from 0.092 to 0.056, and the correlation coefficient increases from 0.835 to 0.936. However, because the MISR RA currently has no fine-mode dust analogs, the amount of value one can extract from these statistics is limited.



A comparison of MISR 550 nm over-land retrieved SSA and AERONET 550 nm SSA is also presented in Figure 5 (panels g-i). At first glance, MISR retrieved SSA does not appear nearly as robust (or as well correlated) as retrieved FMF and non-spherical fraction. For AOD greater than 1.5, RMSE is 0.019 and MAE is 0.013, whereas the correlation coefficient is 0.807. Given that AERONET uncertainty for SSA at these higher AODs is likely in the range of 0.01-0.02 (*Sinyuk et al.,* 2020), it is also possible that AERONET SSA uncertainty may be propagating into our reported statistics (unless the errors for both MISR and AERONET are positively correlated).

### 3.2 MISR RA Over-Water Validation using AERONET

We use the same temporal constraints for our over-water AERONET comparison as were used over land. We apply the following series of tests to help identify good-quality retrievals (for all 48x48 MISR RA retrievals centered on an AERONET station). Quality flags are set for each test.

1. MISR surface height (from DEM) is within 200m of the given AERONET station height
2. At least 7 of 9 MISR cameras valid radiance data
3. MISR pixel must be masked as water
4. MISR retrieved surface cost function < 1
5. MISR retrieved surface AOD < 9
6. NDVI (minimized over all 9 cameras) of MISR reflectances < -0.075
7. (MISR prescribed surface AOD – MISR retrieved surface AOD) < (0.25 * MISR retrieved surface AOD + 0.05)
8. MISR retrieved surface AOD standard deviation among all QA pixels < 2

As for our over-land results, quality flag 1 ensures that we compare pixels at roughly the same elevation to each other and is only used when comparing AERONET AOD to MISR retrieved AOD. Quality flag 2 ensures that a retrieval has enough "good" input data to give high-quality output, which is especially important over-water where up to four cameras could be glint contaminated. Quality flag 3 uses our previously computed land/water mask to make sure that a given pixel is water. Quality flag 4 uses the retrieved surface cost function to screen out poor-quality retrievals. Quality flag 5 screens out pixels with a retrieved surface AOD > 9 (likely cloud), and quality flag 6 will mask cloud out clouds and ephemeral waterways that are not currently water covered. Quality flag 7 is used to identify clouds that have not been screened by the other quality filters. Because the over-water retrieved surface aerosol retrieval does not suffer from the same dramatic loss of sensitivity to AOD seen by the over-land retrieval, the prescribed surface and retrieved surface retrieval values should be similar except in the presence of clouds or over very bright waters. As such, quality flag 7 will also likely eliminate many retrievals over bright waters. Quality flag 8 attempts to remove stray clouds via a large-scale variability filter.





### 3.2.1 AERONET Direct Sun Validation of MISR Over-Water RA

As in our over-land comparison, we apply the flags listed above and require at least 10 quality-assessed retrievals (pixels) for each AERONET coincidence, otherwise the spatially averaged MISR results are not included in the statistics. AOD and ANG statistics for the 4590 MISR quality assessed/AERONET coincidences are shown Table 5.

**Table 5: MISR RA vs. AERONET direct-sun statistics over-water**

| AOD Comparison | # | RMSE | MAE | bias | r |
|---|---|---|---|---|---|
| Retrieved Surface (RS) | 4590 | 0.064 | 0.024 | 0.013 | 0.934 |
| Prescribed Surface (PS) | 4590 | 0.080 | 0.039 | 0.042 | 0.933 |
| Combined Surface (CS) | 4590 | 0.063 | 0.024 | 0.014 | 0.939 |
| | | | | | |
| ANG Comparison (CS Only) | # | RMSE | MAE | bias | r |
| CS ANG I CS AOD>0.05 | 4307 | 0.390 | 0.257 | -0.064 | 0.666 |
| CS ANG I CS AOD>0.20 | 1361 | 0.325 | 0.218 | -0.019 | 0.812 |
| CS ANG I CS AOD>0.50 | 211 | 0.297 | 0.210 | 0.073 | 0.881 |
| CS ANG I CS AOD>1.0 | 25 | 0.272 | 0.201 | 0.136 | 0.910 |

**Same as Table 3, except for MISR RA over-water.**

Figure 6, the over-land equivalent to Figure 2, presents the comparison of MISR over-water AOD for all three retrieval types (retrieved surface, prescribed surface, and combined surface) as both a scatterplot in linear space (to

10 emphasize higher AOD results) and as a log-log 2-d histogram (to compare lower AOD results). Specifically, the MISR retrieved surface algorithm does not suffer from the same level of degradation in results as AERONET AOD increases (there is still a small low bias), which is why we use the retrieved surface algorithm to identify the bounds for the combined surface algorithm over water. Compared to Figure 2 and Table 3, results appear much more consistent with AERONET AOD over-water than over-land, with a combined surface RMSE of 0.063 over-water vs 0.091 over-land, MAE of 0.024 over-water vs

15 0.035 over-land and correlation coefficient of 0.939 over-water vs 0.935 over-land. Although there is little improvement in the total statistics between the retrieved and combined over-water surface results, this may be due to the very limited number of MISR over-water/AERONET coincidences when AOD is elevated (>1; 25 over-water vs 127 over-land). As such (and to be consistent with the previous section), ANG and particle property results are presented subsequently only for the combined surface retrieval.



Figure 6) Same as Figure 2, except for the MISR over-water retrieval.

Figure 7a shows the MISR combined over-water AOD compared to AERONET AOD, with colored rectangular boxes to indicate the retrieval regime of the MISR combined retrieval. Figure 7b shows a plot of |MISR-AERONET| 68th percentile errors as a function of MISR combined over-water AOD. The line fits very well to (0.2 * MISR AOD + 0.01) for all range of retrieved AOD, indicating that this should be a good estimate of expected error.



**Figure 7) Same as Figure 3, except for the MISR over-water retrieval.**

Figure 8 shows the comparison of MISR over-water combined ANG with AERONET ANG as a 2-d histogram for the same AOD bins presented in Figure 5: a) MISR AOD >0.05, b) MISR AOD >0.2, c) MISR AOD>0.50, and d) MISR





AOD > 1. It is clear from Figure 8 that the MISR over-water retrieval algorithm suffers from a small low bias in ANG for pollution/smoke aerosol when AOD is low (<0.2), and this is also represented in the statistics found in Table 5. As expected, the statistics for the MISR over-water retrieval appear better than the over-land results for every AOD regime. Compared to the over-land results for MISR retrieved AOD > 0.20, RMSE is 0.325 vs 0.4 over-water, MAE is 0.218 vs 0.279, and the

correlation coefficient is 0.812 vs 0.717, suggesting that the MISR over-water retrieval has good sensitivity to retrievals of spectral AOD (ANG is derived from this) when AOD is >0.20.

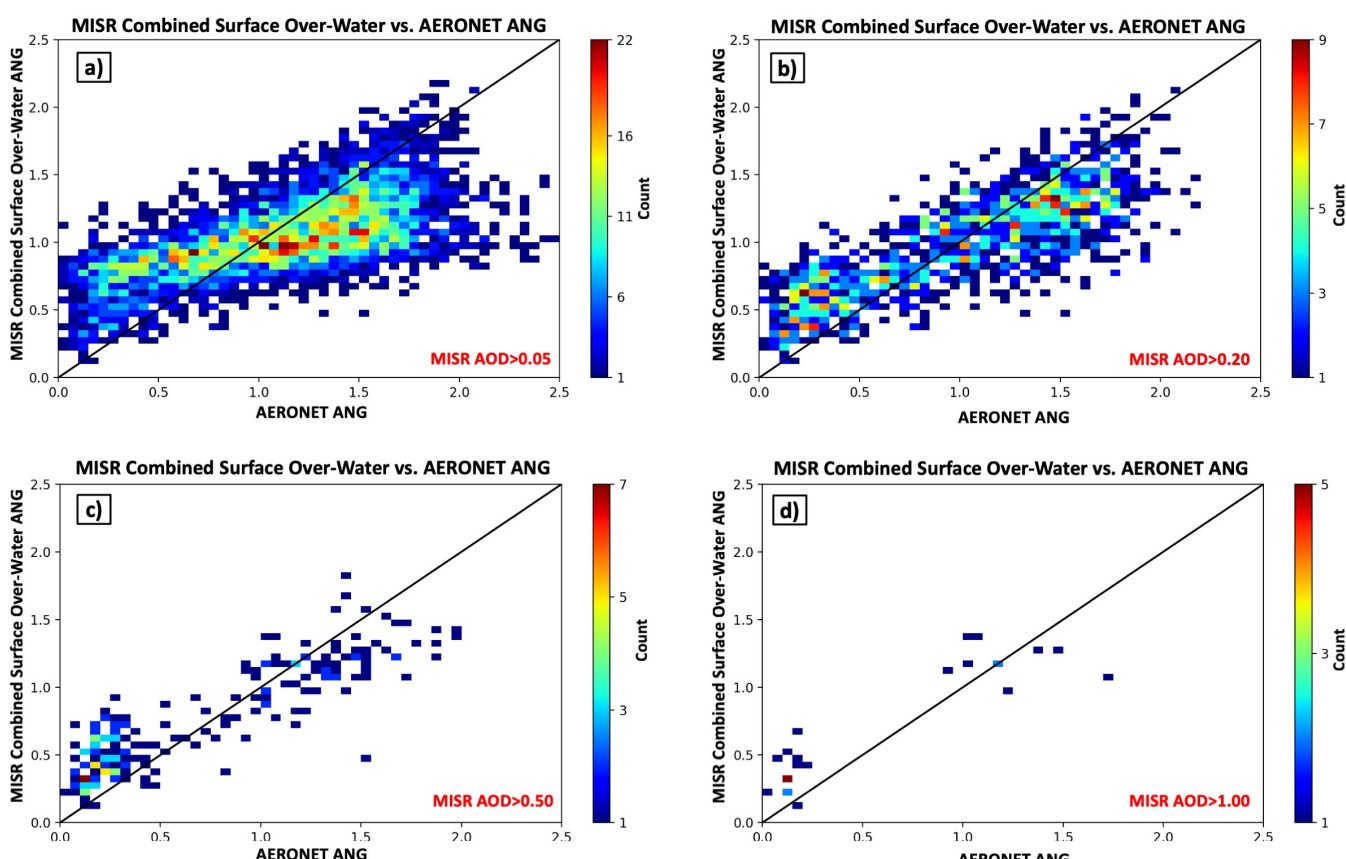

**Figure 8) Same as Figure 4, except for the MISR over-water retrieval.**

### 3.2.2 AERONET Inversion Validation of MISR Over-Water RA

As in our over-land comparison, we use the MISR combined retrieval results with a 4-hour averaging window and 10-pixel minimum, which yields the following number of MISR over-water/AERONET inversion coincidences: 1101 coincidences with MISR AOD > 0.2, 184 coincidences with AOD > 0.5, and 20 coincidences with AOD > 1. Statistics for the MISR over-water vs AERONET inversion comparison are shown in Table 6 and Figure 11 for 550 nm fine-mode fraction, non-spherical fraction due to coarse-mode aerosol, and 550 nm single scattering albedo. Due to the limited number of MISR over-

water/AERONET coincidences with AOD > 1.0, the conclusions one can draw from this dataset will also be limited.





Panels 9a-c show scatterplots of MISR over-water FMF compared to AERONET FMF for AOD ranges listed above. MISR over-water FMF statistics are better than the over-land results for MISR retrieved AOD < 1, especially for the AOD range of 0.2 – 0.5. In this range, over-water vs over-land statistics are as follows: RMSE is 0.159 vs 0.194, MAE is 0.098 vs 0.132, and the correlation coefficient is 0.739 vs 0.597. Interestingly, MISR over-water FMF statistics deteriorate

5 from the 0.5-1.0 retrieved AOD regime compared to the 0.2-0.50 regime, with increases in both RMSE and MAE, even though correlation improves (from 0.739 to 0.830).

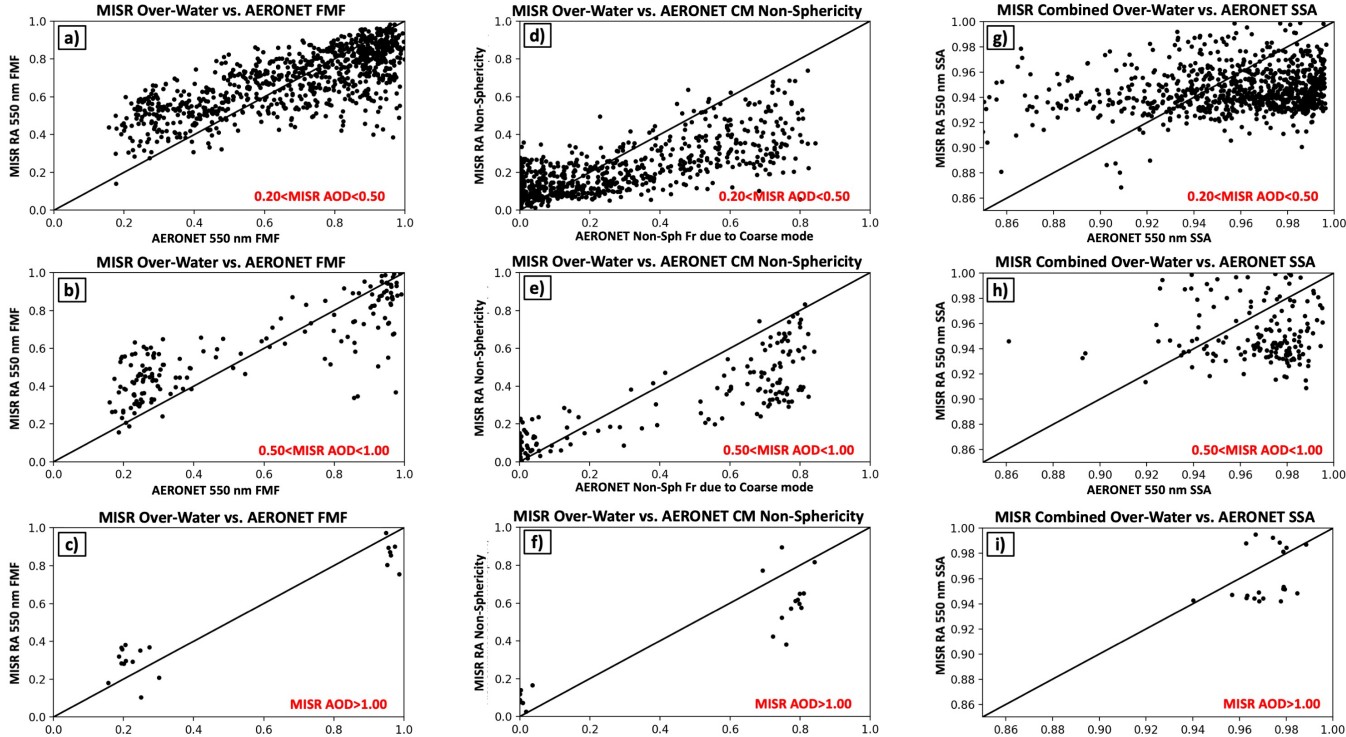

**Figure 9) Same as Figure 5, except for the MISR over-water retrieval. Note the different AOD (compared to Figure 5) bounds embedded in red.**

Panels 9d-f show scatterplots of MISR over-water non-spherical fraction compared to AERONET non-spherical fraction due to the coarse mode. As with FMF, over-water statistics appear more favorable than over-land statistics until retrieved AOD approaches ~1. The same decrease in over-water case numbers and deterioration in statistics (except correlation) is seen as AOD increases from the lowest bin (0.2-0.5) to the middle bin (0.5-1.0). Even though the statistics

15 improve for non-spherical fraction from the middle bin (0.5-1.0) to the highest bin (>1), RMSE and MAE are higher for AOD >1.0 than for 0.2<AOD<0.5. Looking more closely at panels 9a-e, it is also clear that there is significantly more dust present as AOD increases. The statistics bear this out, with AERONET non-sphericity due to the coarse mode increasing from 0.22 at 0.2<MISR AOD<0.5, to 0.425 at 0.5<MISR AOD <1.0, increasing further to 0.507 for MISR AOD>1. Although this effect is also seen in our over-land comparison, the shift is much less dramatic: over-land AERONET non-





sphericity due to the coarse mode is 0.215 for 0.2<MISR AOD<0.5, increasing to 0.282 for 0.5<MISR AOD<1.0, and increasing again to 0.302 for MISR AOD >1. All prior figures involving ANG, FMF, or non-sphericity (over-land and over-water) point to dust retrieval problems when AOD is <1, even though the algorithm will correctly identify pollution or smoke as fine-mode-dominated at lower AOD.

Panels 9g-I show scatterplots of MISR over-water 550 nm single-scattering albedo compared to retrievals of AERONET single-scattering albedo (with the same AOD ranges as above). Although the correlation is quite a bit lower than the results over-land, RMSE and MAE are generally similar between the land and water algorithms within each AOD range. For instance, for over-water AOD>1 (n=20), we report an RMSE of 0.022, a MAE of 0.021, and a correlation coefficient of 0.324, whereas for the 1.0-1.5 AOD bin over-land retrieval (n=62) we report an RMSE of 0.026, MAE of 0.017 and

correlation coefficient of 0.430. Some of the discrepancy between the over-land and over-water results may be due to aerosol regional variability rather than an error in the retrieval itself. Additionally, even though one would expect significantly better constraints on aerosol size from the MISR over-water retrieval at low AOD compared to the MISR over-land retrieval due to the respective surface retrievals, this may not be the case when aerosol loading is elevated, where we use a prescribed surface. At high AOD, the prescribed surface over-land algorithm can make use of all MISR cameras to retrieve information

about aerosol size and absorption, including the information-content-rich forward scattering views when observations are away from the solar equator, whereas these forward scattering views will generally be contaminated by sun-glint and are therefore less usable over water.

**Table 6: MISR RA vs. AERONET almucantar inversion statistics over-water**

| **550 nm FMF Comparison** | **#** | **RMSE** | **MAE** | **bias** | **r** |
|---|---|---|---|---|---|
| FMF l 0.20>AOD>0.50 | 917 | 0.159 | 0.098 | 0.003 | 0.739 |
| FMF l 0.50>AOD>1.00 | 164 | 0.184 | 0.107 | 0.048 | 0.830 |
| FMF l AOD>1.00 | 20 | 0.120 | 0.094 | 0.014 | 0.956 |
| | | | | | |
| **Non-Sph. Fr. Comparison** | **#** | **RMSE** | **MAE** | **bias** | **r** |
| Non-Sph. Fr. l 0.20>AOD>0.50 | 917 | 0.173 | 0.100 | -0.033 | 0.753 |
| Non-Sph. Fr. l 0.50>AOD>1.00 | 164 | 0.218 | 0.127 | -0.112 | 0.852 |
| Non-Sph. Fr. l AOD>1.00 | 20 | 0.176 | 0.149 | 0.068 | 0.911 |
| | | | | | |
| **550 nm SSA Comparison** | **#** | **RMSE** | **MAE** | **bias** | **r** |
| SSA l 0.20>AOD>0.50 | 917 | 0.042 | 0.029 | -0.009 | 0.198 |
| SSA l 0.50>AOD>1.00 | 164 | 0.040 | 0.030 | -0.014 | -0.020 |
| SSA l AOD>1.00 | 20 | 0.022 | 0.021 | -0.010 | 0.324 |

**Same as Table 4, except for MISR RA vs AERONET inversion statistics over-water. All MISR data corresponds to the combined surface retrieval. Note that AERONET inversion results are not ground truth, they represent retrieval results. The AERONET**



**team cautions against the use of results when blue-band AOD <0.4, so comparisons for green band AOD <0.50 should be considered qualitative rather than quantitative.**

## 4 Conclusions

In Limbacher and Kahn [2019], we demonstrated the MISR RA's ability to retrieve AOD and Ångström exponent over ice-
5    free water of any color (turbid, shallow, eutrophic, etc.). Using the same dataset we used in that study, we develop, test, and
present a new version of the MISR RA capable of retrieving aerosol and surface properties over both desert-free land and
ice-free water. We also test the approach of imposing a prescribed surface reflectance at higher AOD, using MODIS MAIAC
RTLS 8-day surface reflectance kernels over land and generic values over water. In addition to validating AOD and
Ångström exponent, we dig more deeply into this dataset by evaluating retrieved fine-mode fraction (FMF), retrieved non-
10   spherical fraction due to coarse mode aerosol (Non-Sph Fr), and retrieved single-scattering albedo (SSA; all parameters at
550 nm).

Over land, using our combined surface approach, the dataset yields 9680 quality-assessed MISR/AERONET direct-
sun coincidences. The MISR RA over-land 550 nm AOD is highly correlated with AERONET 550 nm AOD (r=0.933). The
error statistics are also quite favorable, with an RMSE of 0.091, median-absolute error (MAE) of 0.035, and a small bias of -
0.006. Constraining MISR RA retrieved AOD errors by MISR RA retrieved AOD, we identify a prognostic pixel-level
MISR RA over-land AOD uncertainty of ± (0.225*[MISR AOD] + 0.025), which holds true even when AOD exceeds unity
over-land, unlike for the MISR operational standard algorithm (SA; which suffers from extreme biases in this regime). For
the 565 MISR/AERONET direct-sun coincidences with MISR-retrieved AOD greater than 0.50, we report the following
Ångström exponent statistics: RMSE is 0.359, MAE is 0.250, the bias is 0.171, and the correlation coefficient is 0.852. The
AERONET almucantar inversion dataset yields 505 quality assessed MISR/AERONET coincidences with MISR retrieved
AOD > 0.50 and 107 coincidences with MISR retrieved AOD > 1. For 1.0<MISR AOD<1.5, we report FMF RMSE of 0.107
and FMF r=0.968, Non-Sph Fr. due to the coarse mode RMSE of 0.149 and r =0.936, and SSA RMSE of 0.026 and r=0.43.
All statistics over-land continue to improve for AOD>1.5. Taken together with the Ångström exponent statistics, the over-
land MISR RA yields some qualitative information about aerosol size (FMF and ANG) if retrieved AOD exceeds 0.2, with
excellent quantitative comparison to AERONET beginning at an AOD ~ 1.0. Qualitative retrievals of non-spherical fraction
due to coarse mode aerosol can be made at slightly higher AOD (~0.2-0.5) than is needed to get constraints on size, with
excellent quantitative comparison at an AOD of 1. Depending on retrieval conditions, qualitative retrieval of SSA can be
made at an AOD ranging from 0.5-1.0, with quantitative results (RMSE < 0.02) apparent when AOD exceeds 1.5. Overall,
we note that our assessment of retrieved particle properties from the MISR RA is consistent with the study performed by
Kahn and Gaitley [2015] using the previous version (V22) of the MISR operational aerosol product. However, that work was
limited to AOD < 0.6, as the MISR SA suffers from systematic biases in AOD above this. For the first time, partly because
the MISR RA prescribed surface algorithm allows us to perform aerosol retrievals accurately at much higher AOD, we can



extend their qualitative conclusions about MISR retrieved aerosol type into a more quantitative over-land comparison with AERONET.

Over water our combined surface approach yields 4590 MISR quality-assessed/AERONET direct-sun coincidences. As with the over-land retrieval, over-water AOD is highly correlated (r=0.939) with AERONET 550 nm AOD. Error

statistics also improve, with an RMSE of 0.063, MAE of 0.024, and a small bias of 0.014. Prognostic pixel-level AOD error improves slightly to ± (0.20*[MISR AOD] + 0.01). For the 211 MISR/AERONET direct-sun coincidences with MISR-retrieved AOD greater than 0.50, we report the following Ångström exponent statistics: RMSE is 0.297, MAE is 0.210, the bias is 0.073, and the correlation coefficient is 0.881. The AERONET almucantar inversion dataset yields 184 quality-assessed MISR/AERONET coincidences with MISR retrieved AOD > 0.50 and 20 coincidences with MISR retrieved AOD

> 1, which greatly limits our ability to draw conclusions about retrieved aerosol particle properties over-water. For MISR AOD>1.0, we report FMF RMSE of 0.120 and FMF r=0.956, Non-Sph Fr. due to the coarse mode RMSE of 0.176 and r =0.911, and SSA RMSE of 0.022 and r=0.324. Qualitative retrievals of aerosol type appear similar to the over-land retrieval, with the expectation that crude constraints on aerosol size can be made at lower AOD. Based on the biases present in MISR retrieved FMF, non-sphericity, and Ångström exponent (over both land and water), it appears likely that the inclusion of a

fine-mode transported dust analog will improve comparisons against AERONET, especially for comparisons of FMF and sphericity.

This paper represents the first iteration of the combined MISR RA over-land + over-water retrieval. The authors plan to use to results of this study to further refine the aerosol particle properties used by the algorithm and improve our characterization of the surface used by the prescribed surface algorithm over both land and water. In the future, we will

likely include all AERONET direct-sun/inversion coincidences with MISR for the entire 22-year data record rather than the 4 years that were included here, as this will improve our ability to draw conclusions about aerosol particle properties, especially over water.

**Data availability.**

All MISR RA validation data used for this manuscript will be published to the NASA Langley DAAC prior to publication.

**Author contributions.**

Originally developed by RAK, the MISR RA has been a joint effort of JAL and RAK since early 2011. The updated algorithm presented here was developed by JAL (with supervision by RAK), while the manuscript was produced with input from both JAL and RAK. JL developed the dust aerosol model used for this manuscript.

**Competing interests.**

The authors declare that that have no conflict of interest.



**Acknowledgments**

We thank our colleagues on the Jet Propulsion Laboratory's MISR instrument team and at the NASA Langley Research Center's Atmospheric Sciences Data Center for their roles in producing the MISR Standard data sets, and Brent Holben at NASA Goddard and the AERONET team for producing and maintaining this critical validation dataset. We thank Alexei

Lyapustin and the MAIAC team for the MODIS MAIAC products used in this manuscript. We also thank Drs. Rozanov and the SCIATRAN team for their work on the SCIATRAN product. CCMP Version-2.0 vector wind analyses are produced by Remote Sensing Systems. Data are available at [www.remss.com](www.remss.com). This research is supported in part by NASA's Climate and Radiation Research and Analysis Program under Hal Maring, NASA's Atmospheric Composition Program under Richard Eckman, and the NASA EOS MISR and Terra projects.

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
