# Peer review of "The New MISR Research Aerosol Retrieval Algorithm: A Multi-Angle, Multi-Spectral, Bounded-Variable Least Squares Retrieval of Aerosol Particle Properties over Both Land and Water"

_Atmospheric Measurement Techniques, 2022_

## Referee Comment (RC1)

Review of paper:

**A new MISR research aerosol retrieval algorithm: a multi-angle, multispectral, bounded-variable least square retrieval of particles properties over both land and water validation by J. Limbacher et al.**

**Highlights**

- very detailed MISR retrieval approach of the research algorithm (RA)
- evaluations not just limited to AOD
- nice illustrations of strength and weaknesses

**Concerns**

- comparisons of results to the older RA and also to the standard (SA) algorithm
- likely incorrect assumptions about dust bias solutions
- regional (Sahara, off-Sahara) testbed cases are missing to examine the dust problem

**General comments**

MISR comprises a set of multi-spectral sensors oriented into different directions. Data combination in developed retrievals are quite powerful to determine multiple aerosol properties (at cloud-free conditions) with accuracies usually not matched by other satellite sensors. In addition, the long-term data record (in operation since 2000) makes this data-set highly attractive.

While a new retrieval is suggested, the 'research' aspect makes me frown, as new efforts for retrievals will only count for data-users, if retrievals are applied to the entire data-record (e.g. for climate studies) and not to a limited number of cases. In any case, statistical comparisons of retrievals (newRA vs oldRA vs stdR) for a limited period could be a nice addition to demonstrate newRA capabilities - in the discussion section.

The paper is rather technical and introduces a new (less complex) aerosol model with the number of permitted aerosol types – compared to the standard algorithm - reduced to 17. This is a step in the right direction (also for unique answers). Still I suggest further changes. I question the necessity of 'very small' (reff ~0.06um) sizes, which are hardly contributing to optics – unless they are very absorbing. However, 'very small' BC is quickly increased in size to 'small' and even to 'median' sizes as (absorbing OC and scattering SU) condensate attaches. Hereby for OC a weak absorption in the mid-vis but a strong absorption towards the UV should be assumed so that a BC(core, reff~0.06um))/OC(shell) type can mimic 'brown carbon'. Thus, there is high potential to reduce the number fine-mode choices. On the other hand, the coarse-mode choices are far too simple as also larger mineral dust sizes (with lower mid-vid SSA for the same Rfimag) should be considered. I am almost certain, that this will reduce FMF, non-sph and ANG biases of this new MISR retrieval.

Otherwise, this is a very informative paper.

P.S. I have attached a summary of the top-down concepts of my MACv3 climatology, where is the coarse-mode AAOD information – along with the dust coarse-mode AOD is used to determine coarse dust AOD and coarse dust size.

(note, fine-mode dust AOD and fine-mode SSA contributions are considered secondary and being considered conservative scatters, they are attributed to non-absorbing fine-mode 'SU' in the MAC climatology)

**Specific comments**

5/4 to table 1: I missed a couple of (at least one) larger dust sizes (as with larger dust AOD usually also the mineral dust sizes are larger ... which strongly adds [coarse mode] absorption). I also would get rid of the very small aerosol types and would start with 'small'. Here I would add a mixture (a 'very small' BC core size with an organic OC shell to yield a 'small' mix type) as mostly (or only) scattering usually quickly condenses on BC. Hereby I would also define organic ('OC') with a strong increasing absorption increase towards the UV and a pure scattering fine-mode ('SU') component for both 'small' and 'medium'). Hereby the BC(core)/OC(shell) type covers the artifical 'brown carbon' component. This reduces the minimum number of types to be considered to eight: small: BC/OC, SU, OC / medium: OC, SU / large: SS and DU / very large: DU For the content in the table I would like to see next to the eff. radius also assumed distribution width information (std.dev or variance) rather than r.min and r.max. In addition, for the Angstrom parameter, the defining wavelengths need to be listed in the captions and SSA data should be shown that actually relate to the type assumption for size and composition (and not just made up by an arbitrary value, like 0.8). This will also help later to improve to relate types to those used in global modeling. Finally for AAE (as fro Angstrom) the defining wavelengths are needed or simply add an SSA value at another relevant wavelength (e.g. 440nm) - based on RFimag spectral data for the particular size of that type.

6/6 one large non-spherical model is not sufficient - especially over the Sahara and for Saharan outflow - where extra large dust-sizes, if ignored, likely cause AODc underestimates and also possibly absorbing fine-mode overestimates

7/10 why not using initial values from a climatology or data from the most recent retrieval at that location?

9/7 actually it would be great if this mixture information could be saved – at least for a couple of test-cases. In this way I could be explored to what degree each of the now 17 types contributes also in efforts to reduce the number of required types.

9/25 if you remove the smallest (re=0.06um) sizes (then you are down from 9 to 7) and when you add a larger non-sph dust (e.g. re ~ 5um) then you are up from seven to 8)

11/12 I assume that in the combined surface/aerosol retrieval constraints are built in, which do not allow for negative albedos or negative AOD.

18/24 Nice, that the comparison are shown for both the largest and (via ln/ln) the smaller AODs

I think, that the missing coarse mode (e.g. MISR Angstrom overestimates at smaller AERONET Angstrom) has much to do with the missing larger coarse-mode sizes. This would have been also apparent, if the deviations of the scatter plot would have been places as a function of location. (This behavior is similar to SLSTR biases, which also only consider one dust size in their model and then attribute absorption to the fine mode (AODf overestimates), while this absorption should have gone to dust size (increases))

22/10 For me the fine-mode non-spherical dust is not the issue. If absorption is allowed not only to be associated with fine mode (BC, OC) and relative small DU, but also with larger DU size, then most of the MISR biases now will go away (better non-sphere, better fine-mode, better SSA) in comparisons to DUST cases (on the other hand at the largest AODs AERONET actually gives size-distributions, although size with re>10um are likely missed by these inversions). In the scatter plot presentation the 0.5-1.0 AOD already relates to the largest AOD events, so I consider only the first row in figures 5 and 9 relevant. (rows 2 and 3 in Figure 5 and row 3 in Figure 9 are interesting but less meaningful because of much lower statistics are less meaningful). Why not showing in Figure 5 the same range statistics as in Figure 9?

AOD >0.4 is already a large AOD and AOD>1 are very rare, so I suggest to focus on the 0.3 to 1.0 range for coarse mode AOD. And the scatter plots for FMF, non-sph and SSA do not indicate co-locations ... so I would look at dust outflow off Africa (ocean) and dust over the Sahara (land) to investigate the dust retrieval problem.

Attached, below is the description of the top-down approach of the MAC aerosol climatology. In the top-down approach also the dust size is retrieved (from the coarse-mode AAOD). In the third column for Figure A3 the extracted dust effective radius (divided by 10) is illustrated. Especially over the Sahara and for dust outflow ontoe Atlantic – especially during JJA due dust size are significantly larger ... even for monthly averages (of this climatology).

**MAC v3 details**

The Max-Planck Aerosol Climatology (MAC) offers merged monthly global maps for aerosol optical properties. In the merging process, multi-annual observational statistics of photometry from the ground is forced on spatial context supplied aerosol component 'bottom-up' modeling. The merged aerosol optical properties focus on aerosol column amount and aerosol column absorption, separately for smaller 'fine-mode' aerosol and larger 'coarse-mode' aerosols. In Figure A1, multi-annual averages from photometry (AERONET/MAN with absorption data only over continents) are compared to multi-(AeroCom phase 3) model interquartile averages (AC3-iqa). In addition, in that Figure A1 the resulting MAC version 3 maps are presented along with applied regional % changes to AC3-iqa.

---

## Referee Comment (RC5)

[referee-annotated manuscript omitted]

---

## Author Response (AR1)

**Reviewer 1-2**

**Review of paper:**

- A new MISR research aerosol retrieval algorithm: a multi-angle, multispectral, bounded-variable least 5 square retrieval of particles properties over both land and water validation by J. Limbacher et al. Highlights
  - very detailed MISR retrieval approach of the research algorithm (RA)
  - evaluations not just limited to AOD
- 10 nice illustrations of strength and weaknesses

**Concerns**

- comparisons of results to the older RA and also to the standard (SA) algorithm
- likely incorrect assumptions about dust bias solutions
- regional (Sahara, off-Sahara) testbed cases are missing to examine the dust problem
- 15

**General comments**

MISR comprises a set of multi-spectral sensors oriented into different directions. Data combination in developed retrievals are quite powerful to determine multiple aerosol properties (at cloud-free conditions) with accuracies usually not matched by other satellite sensors. In addition, the long-term data record (in operation since 2000) makes this data-set highly attractive.

20

While a new retrieval is suggested, the 'research' aspect makes me frown, as new efforts for retrievals will only count for data-users, if retrievals are applied to the entire data-record (e.g. for climate studies) and not to a limited number of cases. In any case, statistical comparisons of retrievals (newRA vs oldRA vs stdR) for a limited period could be a nice addition to demonstrate newRA capabilities - in the discussion section.

- 25 The paper is rather technical and introduces a new (less complex) aerosol model with the number of permitted aerosol types - compared to the standard algorithm - reduced to 17. This is a step in the right direction (also for unique answers). Still I suggest further changes. I question the necessity of 'very small' (reff~0.06um) sizes, which are hardly contributing to optics - unless they are very absorbing. However, 'very small' BC is quickly increased in size to 'small' and even to 'median' sizes as (absorbing OC and scattering SU) condensate attaches. Hereby for OC a weak absorption in the
- 30 mid-vis but a strong absorption towards the UV should be assumed so that a BC(core, reff~0.06um))/OC(shell) type can mimic 'brown carbon'. Thus, there is high potential to reduce the number fine-mode choices. On the other hand, the coarsemode choices are far too simple as also larger mineral dust sizes (with lower mid-vid SSA for the same Rfimag) should be considered. I am almost certain, that this will reduce FMF, non-sph and ANG biases of this new MISR retrieval. Otherwise, this is a very informative paper.

35 The authors thank Stefan Kinne for his thorough comments. A couple broad comments are addressed below

- 1. The algorithm described here represents the first version of the algorithm to be used to generate a global MISR Research Product which we are working toward producing for all 22+ years of MISR data. (We just recently received funding for the first time to produce a Research Product, largely based on the work presented in this and other recent MISR RA papers.)
- 40 2. This algorithm represents the first RA over-land results that the lead author has published (so no previous RA version to compare to).
  - 3. We agree with your comments about the 0.06 micron aerosol models and have removed them. We have also added a spherical non-absorbing 0.57 micron effective radius particle as additional coarse mode aerosol mode. This change has been on our list to do since Kahn et al. (2010). We have also replaced the 1.28 micron effective radius analog with a 2.8 micron effective radius analog, as the algorithm has less sensitivity to size differences
- 45 in the "coarse" range, but will be much more likely to distinguish between the 0.57 and 2.8 micron analogs.

- 4. We have added new non-spherical models with the same size distributions as our spherical analogs. The nonspherical analogs use the same set of refractive indices (although different than for our spherical analogs), which results in much more absorption for the larger size distributions.
- 5. The RA is also a testbed for changes to the SA, so some of the advances presented here might appear in the SA before a full Research Product is completed.

P.S. I have attached a summary of the top-down concepts of my MACv3 climatology, where is the coarse-mode AAOD information – along with the dust coarse-mode AOD is used to determine coarse dust AOD and coarse dust size.

10 (note, fine-mode dust AOD and fine-mode SSA contributions are considered secondary and being considered conservative scatters, they are attributed to non-absorbing fine-mode 'SU' in the MAC climatology)

**Specific comments**

5

- 5/4 to table 1: I missed a couple of (at least one) larger dust sizes (as with larger dust AOD usually also the mineral dust sizes are larger ... which strongly adds [coarse mode] absorption). I also would get rid of the very small aerosol types and would start with 'small'. Here I would add a mixture (a 'very small' BC core size with an organic OC shell to yield a 'small' mix type) as mostly (or only) scattering usually quickly condenses on BC. Hereby I would also define organic ('OC') with a strong increasing absorption increase towards the UV and a pure scattering fine-mode ('SU') component for both 'small' and 'medium'). Hereby the BC(core)/OC(shell) type covers the artifical 'brown carbon' component. This reduces the
- 20 minimum number of types to be considered to eight: small: BC/OC, SU, OC / medium: OC, SU / large: SS and DU / very large: DU

We are adding 2 non-spherical fine-mode non-spherical models (0.12, 0.26 micron effective radius), 3 coarse mode dust models (0.57, 1.28, and 2.8 micron effective radius) and 1 more spherical non-absorbing analog (2.8 micron effective radius). This allows us to retrieve total sphericity (rather than only coarse-mode sphericity) and coarse-

- 25 mode size. As we pointed out above, we are removing the smallest aerosol models (0.06) and adding a 0.57 micron effective radius "coarse" mode spherical analog. This gives us a total of 17 discrete aerosol models, but only adds one more retrieved parameter. These choices are motivated by our growing experience with the particle-property distinctions we can make with MISR under good retrieval conditions.
- 30 For the content in the table I would like to see next to the eff. radius also assumed distribution width information (std.dev or variance) rather than r,min and r,max. In addition, for the Angstrom parameter, the defining wavelengths need to be listed in the captions and SSA data should be shown that actually relate to the type assumption for size and composition (and not just made up by an arbitrary value, like 0.8). This will also help later to improve to relate types to those used in global modeling. Finally for AAE (as fro Angstrom) the defining wavelengths are needed or simply add an SSA value at another relevant wavelength (e.g. 440nm) based on RFimag spectral data for the particular size of that type.
- 35 wavelength (e.g. 440nm) based on RFimag spectral data for the particular size of that type. We added the log-normal characteristic width, and the radius as well. We apologize for not including wavelengths for our Angstrom exponent. We use all four MISR bands for determining this value. Although the 0.8 and 0.9 greenband SSA are rather arbitrary, rather than trying to match a few specific aerosol types that might have been observed in a few circumstances, we are taking the approach of covering a broad range of values that are seen in
- 40 nature. This is also consistent with limited MISR sensitivity to SSA. As the prescribed surface algorithm will interpolate between the different bins, it shouldn't really matter what the bins themselves are (as long as they are densely spaced). The same wavelengths (all 4 MISR bands) are used for AANG as well, we added this).

6/6 one large non-spherical model is not sufficient - especially over the Sahara and for Saharan outflow - where extra large dust-sizes, if ignored, likely cause AODc underestimates and also possibly absorbing fine-mode overestimates

We are adding a second model at 2.8 microns, which will also result in significantly lower blue-band SSA (constant ni for all dust models are used), and we are replacing the 1.28 micron effective radius analog with a 0.57 micron effective radius analog. As the longest MISR wavelength is 0.87 microns, we have limited sensitivity to particle size for particles larger than 2-3 microns.

2

7/10 why not using initial values from a climatology or data from the most recent retrieval at that location? The algorithm is set up to run from a static value globally, in part so our results are not unnecessarily skewed toward our pre-conceived biases (i.e., "confirmation bias," as per *Schutgens et al.*). Of course, we could do this in the future, but the current approach seems to yield reasonable results without introducing this bias.

**5**

9/7 actually it would be great if this mixture information could be saved – at least for a couple of test-cases. In this way I could be explored to what degree each of the now 17 types contributes also in efforts to reduce the number of required types. This information will absolutely be saved for our research product, but for our current manuscript we will continue saving data this way, as this is just an initial validation study.

10

30

9/25 if you remove the smallest (re=0.06um) sizes (then you are down from 9 to 7) and when you add a larger non-sph dust (e.g. re ~ 5um) then you are up from seven to 8) 11/12 I assume that in the combined surface/aerosol retrieval constraints are built in, which do not allow for negative albedos or negative AOD.

We are removing the 0.06 micron effective radius set of analogs (5) and adding a 0.57 micron effective radius non-

- 15 absorbing (and non-spherical analog) as well as a 2.8 micron effective radius non-absorbing (and non-spherical) analog, while removing the 1.28 micron non-spherical analog (which gives us 17 components still). You are correct that we have limits on the retrieved (and prescribed) surface albedos (and Lc as well). These were added in the updated manuscript.
- 20 18/24 Nice, that the comparison are shown for both the largest and (via ln/ln) the smaller AODs **Thanks, we know some people prefer log/log and some people prefer linear scales.**

20 I think, that the missing coarse mode (e.g. MISR Angstrom overestimates at smaller AERONET Angstrom) has much to do with the missing larger coarse-mode sizes. This would have been also apparent, if the deviations of the scatter plot would

25 have been places as a function of location. (This behavior is similar to SLSTR biases, which also only consider one dust size in their model and then attribute absorption to the fine mode (AODf overestimates), while this absorption should have gone to dust size (increases))

Some of this is surely a lack of information content, but we have added: 2 fine-mode dust models, 1 more coarse mode dust model, 1 more coarse-mode spherical model, and removed the 5 0.06 micron effective radius models. The changes made have definitely reduced the bias, although it is still present.

22/10 For me the fine-mode non-spherical dust is not the issue. If absorption is allowed not only to be associated with fine mode (BC, OC) and relative small DU, but also with larger DU size, then most of the MISR biases now will go away (better non-sphere, better fine-mode, better SSA) in comparisons to DUST cases (on the other hand at the largest AODs AERONET

- 35 actually gives size-distributions, although size with re>10um are likely missed by these inversions). In the scatter plot presentation the 0.5-1.0 AOD already relates to the largest AOD events, so I consider only the first row in figures 5 and 9 relevant. (rows 2 and 3 in Figure 5 and row 3 in Figure 9 are interesting but less meaningful because of much lower statistics are less meaningful). Why not showing in Figure 5 the same range statistics as in Figure 9?
  Respectfully, AERONET seems to indicate that this is at least partly due to fine-mode dust. We have added fine-
- 40 mode dust models, but have also taken your suggestion and added another coarse-mode dust model as well (2.8 micron effective radius). MISR RA results are clearly improving with AOD (and this is an important point), so we will keep these other AOD ranges as well. Unfortunately, we don't have the numbers of high-AOD cases over-water to plot the results with the same AOD ranges. This is to be expected.
- 45 29/4 AOD >0.4 is already a large AOD and AOD>1 are very rare, so I suggest to focus on the 0.3 to 1.0 range for coarse mode AOD. And the scatter plots for FMF, non-sph and SSA do not indicate co-locations ... so I would look at dust outflow off Africa (ocean) and dust over the Sahara (land) to investigate the dust retrieval problem. Attached, below is the description of the top-down approach of the MAC aerosol climatology. In the top-down approach also the dust size is retrieved (from the coarse-mode AAOD). In the third column for Figure A3 the extracted dust effective radius (divided by
- 50 10) is illustrated. Especially over the Sahara and for dust outflow ontoe Atlantic especially during JJA

AOD from smoke and dust plumes can easily exceed 1 (whether AERONET is well-suited to typically capture such events is a different matter). Considering that aggregate particle property results appear to be worsening while mean retrieved non-spherical fraction is increasing (plus the lack of multiple dust models) seems to indicate a definite problem with our dust retrieval. As we have added another coarse-mode dust model (and several fine mode dust

5 models), we think this should be sufficient for now. We are now retrieving fine-mode size (+ ssa and black-smoke Fr), coarse-mode size, FMF, and total non-sphericity, so we should also be able to retrieve dust effective radius as well.

**CORRECTION to REVIEW**

in my review I also provided the MACv3 aerosol top down aerosol type approach. Unfortunately there is an error in table 1 as the (midvis) SSA for large mineral dust is (not constant) but a function of size (the table lists the same

- table 1 as the (midvis) SSA for large mineral dust is (not constant) but a function of size (the table lists the same values 0.962 value).
   Table 1 correction: for re= 1,5, 2.5, 4.0, 6.5 and 10 the mid-visible dust SSA values are 0.962, 0.931, 0.918, .882 and .840 for the same imaginary part (here 0.0011) ... that is very important to understand my comments of the review. The authors understood what you were trying to convey and believe you will be satisfied with the revised manuscript
- 15 (with new particle models).

20

25

30

35

40

45

**Reviewer 3**

The manuscripted by Limbacher et al provide a thorough and interesting study on the impact of surface reflectance on aerosol retrievals using a new MISR research algorithm. Analyses are conducted by

- 5 using four years of MISR data over both land and water. Details in the aerosol model updates and optimization algorithms are provided, with improvement quantified by comparing quality ensured AROENT data and MISR retrieval results. The MISR algorithm has been well optimized for aerosol retrievals. The new research algorithm further demonstrates the most current capability of aerosol retrievals using multi-angle measurements.
- 10

Specifically, the main motivation of this work is the observation of large biases in retrieved aerosol optical depth (AOD) as aerosol loading increases (>1). To resolve this issue, the authors proposed to use the surface reflectance data from the Multi-Angle Implementation of Atmospheric Correction (MAIAC). A combined algorithm is developed with surface properties directly retrieved for low AOD

15 (<1), and a prescribed surface reflectance from MAIAC for large AOD (>2), and a linear combination of the two surface options are used for 1<AOD<2. By comparing with AERONET product and the MISR research algorithm product, the AOD uncertainties are well quantified as: ± (0.225\*[MISR AOD] + 0.025) over land, and ±(0.20\*[MISR AOD] + 0.01) over water.</li>

**20**

This study provides useful experiences and techniques in exploiting aerosol and surface information from multi-angle measurement. Please find my suggestive comments for the authors to consider. The authors thank Meng Gao for his thorough comments.

**25**

35

Main comments:

Most of my questions are related to how the surface reflectance are treated and how they impact retrieval results:

Since there is a larger number of retrieval parameters when using directly retrieval surface properties, it makes sense that there could be large uncertainties. But it is still not clear to me why this leads to a negative bias of AOD as clearly shown in Fig 2(b).

The retrieved surface AOD is shown in Figures 2a and 2d. The large negative bias when AERONET AOD is  $>\sim$ 1.5 is likely due to the large number of free parameters of the over-land retrieval (19) compared to the over-water retrieval (10), and the algorithm's inability to establish a deep local minimum in cost function when aerosol loading is elevated.

- 2. Page 4, line 25 "The fact that this bias correction was not sufficient to remove the AOD bias seen in the prescribed surface retrieval over-land (especially at AODs < 0.20) indicates that a camera-by-camera correction should probably be used in the future." Since the MAIAC reflectance has been corrected according to MISR retrieval results at low
- AOD (Page 4, since line 19), do you suggest the angular shape is still different between the MAIAC and retrieved surface reflectance? Is this part of the reason to have the bias in AOD retrievals?
   We only compared the MISR retrieved surface albedo to the MAIAC surface albedo for these low AOD cases where the retrieved surface algorithm performed well (as compared to AERONET). We then scaled the surface reflectances for all MISR cameras as described in the manuscript. We never compared camera-to-camera results, as this could result in a significant digression from the main purpose of the current paper. It

**is likely that the angular shape is different between MAIAC and MISR and that this likely contributes to the AOD bias.**

- 3. Since the authors have done retrieval using both retrieved and prescribed surface reflectance, it could be useful to compare the angular/spectral shape of these surface reflectance to understand exactly where the difference are. Specifically:
  - a. What are the retrieved surface reflectance difference under low and high AOD? How do they compare with the prescribed surface reflectance?
  - b. How does the surface reflectance (retrieved and prescribed) impact aerosol property retrievals differently? Currently only AOD are discussed which shows clear bias over land, it would be interesting to understand how the surface reflectance impacts other properties, such as SSA, FMF etc.

These are fantastic suggestions for a future paper on surface reflectance retrieval, but the current paper is already quite long, and the authors want to limit to focus the current manuscript on MISR RA retrieved aerosol loading/aerosol properties. We would also likely need the full set of MISR/AERONET coincidences (with MISR radiances), rather than the ~4 years included in this paper to do this job adequately.

- Page 25, Fig 6, MISR retrieved surface case seem work good over water comparing with the prescribed ocean surface. Does the prescribed ocean surface derived in the same way from MAIAC as discussed for land? Do you have the same correction coefficients applied for the surface reflectance over water? I am curious why there is less AOD bias over ocean than over land.
- This is likely due to the over-water algorithm having significantly fewer free parameters compared to the overland retrieval (10 vs 19), combined with the Lambertian water-leaving reflectance assumption possibly holding in this case better than the shape-similarity assumption used over land. The prescribed surface over water is simply a set of static remote-sensing reflectances (ocean color), with the values presented in the paper. It works pretty well at high-AOD because water color tends to be less variable (especially over open-ocean) than the color/brightness of land surfaces. There was no need for correction coefficients for the over-water retrievals, as we don't use MAIAC.

Minor comments:

5

10

Page 3, line 29: "SSA spectral slope ("Brown Smoke" AOD fraction)". Are they the same here?
 We have changed this to "...SSA, and brown smoke AOD fraction (analogous to SSA spectral slope)..."

Page 4, line 5, "applies a spectrally invariant angular-shape-similarity assumption to derive 5 the surface reflectance (over land)". This is probably explained in later discussions, but do you assume that the same land surface reflectance at different

35 angles and wavelengths? This is explained later, but it simply means that the color of the surface is not allowed to change with view angle, only the brightness (the ratios of surface reflectance are essentially fixed). This is a widely used assumption for multi-angle retrievals over land, and has been shown empirically to be valid in many cases.

- Page 4, line 7, "whereas the other algorithm prescribes the surface reflectance for both land and water from other sources", specify the sources or add reference?
   We have changed this to "...whereas the other algorithm prescribes the surface reflectance for both land (MAIAC) and water (uses a static set of remote-sensing reflectances)."
- 45 Page 4, line 11, "We then correct these TOA reflectances for the following: gas absorption, out-of-band light, stray-light from instrumental artifacts, flat-fielding, and temporal calibration trends". Do you have an estimated accuracy after all those correction in the measurement?

We assume an uncertainty in TOA reflectance of 0.003 + 2% in the algorithm, which seems reasonable given what is known about atmospheric gases and the MISR validation work of Bruegge et al., but we have no way to further verify this

50 this.

Page 4, Line 21/22, "surface reflectance", are they defined in the same way in Eq (1) using ETOA (or EBOA)? The surface reflectance as described here would use EBOA, the irradiance at BOA.

5 Page 5, Line 7, "10m wind-speed". What 10m mean here? The wind speed is retrieved, right? No, and we have added that this is prescribed here.

Page 5, Line 10, "appropriate solar/viewing geometry", do you consider spherical shell effect of the atmosphere? No, the RT code is run in plane-parallel mode. The plane-parallel approximation is adequate for this application in nearly all cases.

Page 6, Table 1, it would clear to explain BrS and BlS in the caption. Thanks for catching this.

15 Page 6, Line 7, how to do you define "non-sphericity" by mixing two coarse modes? The algorithm selected either spherical or non-spherical models, non-sphericity is the fraction of 550nm AOD that occurs due to non-spherical aerosol. This has been updated.

Page 8, line 16, "(2)" and "(3)" seem not used for referred later on?

Page 8, Line 20, cost function seems not normalized by the total number of measurements (N)? The current definition seems agree with a Chi square function which will have the most probable value at N. Is this the case here?

The sum of the weights corresponds to the total number of weighted measurements. Over-land, the sum of w is frequently 36.

Page 9, Line 15 "and MAIAC retrieved surface reflectance error (which should be much larger for the MISR 70Ë -viewing cameras than for the near-nadir cameras). Does this relate to earth spherical shell effect too? There is certainly the possibility of plane-parallel errors at high latitudes, where steep view angles combine with steep

30 sun angles. We have added this as a potential error source, thanks.

Page 10, line 15/18, "set the result to 0", so you are finding both A\* and Lc to minimize the cost function, right? (I appreciate the authors provide details in the optimization approach (eg. Sec 2.1.2). The optimization are represented by a system of linear equations, which seems work well for this algorithm.)

35 Yes, the algorithm retrieves both. However, because the retrieval of Lc depends on the retrieved value of A, these are not really independent retrievals. This interdependence is also a potential source of our AOD bias.

Page 11, line 19, "an additional 9 pieces of information", do you mean the total parameters for land surface are 9+4=13? **Correct**

40

45

10

25

Page 11, line 26, what is the 'prescribed surface AOD', are they also provided by MAIAC? This is the AOD from our prescribed surface algorithm in 2.1.1, we have clarified this.

Page 15, line 12, Do you remove the measurements at particular cameras if the inputs are not 'good'? **Yes, if MISR radiometric QA indicates they are bad, or if reflectance is <0.002.**

Page 15, Line 14, cost function < 1, check the normalization of the cost function as mentioned previously. Because we are summing over weights (which will be 1 for good measurements), our function should be accurate.

20 They are referred to in Figure S1.

Page 15, line 30, "A larger 2nd derivative corresponds to a steeper minimum in our cost function with respect to AOD; we use 10 as a lower bound here in quality flag 6 as this tends to mask out some lower quality results (mostly clouds)". How do you determine the threshold? Since the derivatives are available, can the authors compute the uncertainties using error propagation, which can provide a more meaningful criteria?

5 This threshold (and others) was determined by looking at MISR/AERONET comparisons themselves. This is truly a cloud screening metric. We could take the aggregate derivative for our "effective" aerosol model to come up with a more analytical approach to uncertainty, but we will probably address that in a future paper.

Page 17, line 8, a prognostic error is introduced here, but not well explained. Some information seems scattered in the Fig 3 captain and discussion from later sections. It would be useful to explain early how the error is computed. Another question: what dataset bins are used to compute the 68th percentiles? Are these bins with respect to AOD, reflectance, uncertainty? Fig 3(b): 2% of reflectance?

**Thanks for catching this. We have added the following:**

"This prognostic error is taken as a line fitted to the 68th percentile absolute AOD errors (with respect to AERONET), binned at every 2% of MISR retrieved AOD (so 50 bins in total)."

Page 22, Fig 5: It seems the MISR algorithm have the flexibility to deal with different aerosol types (therefore different refractive index). For large AOD at Fig 5 (bottom row), the data are peaked at either small or large FMF, which results in better SSA and non-sphericity agreement with AERONT. But for small AOD, there are many intermediate FMF values. If I

20 recall correctly, AERONET retrieval algorithm assumes the same refractive indices for both fine and coarse mode. So the AERONET product should have better representation for fine or coarse mode dominated cases. Does this partially explain what we observe in Fig 5 here?

We believe this indicates at higher AOD we are more likely to see homogenized aerosol types (mostly dust or mostly smoke, for instance). The authors agree that errors in fine/coarse mode assumptions when the refractive indices for

25 two modes in the atmospheric column are different could lead to errors in AERONET particle properties, especially if the plume was not either fine-or-coarse dominated though.

30

35

40

45

**Reviewer 4**

This manuscript by Limbacher et al. present a new MISR research algorithm (RA) for retrieving aerosol over land and water surface. To address the issues of large biases of high aerosol loading in MISR

- 5 operational standard aerosol algorithm (SA), the proposed RA utilized and combined 2 schemes: (i) retrieved surface; (ii) prescribed surface from MODIS/MAIAC product. If the prescribed surface algorithm reported AOD<1, then the results from retrieved surface algorithm will be used; if the prescribed surface algorithm reported AOD>2, the results will adopt from prescribed surface algorithm; while if the 1<AOD<2, the results will be merged from 2 algorithms. In general, the methodology is
- 10 sound. The validation with AERONET suggest a good quality for AOD, ANG as well as FMF, SSA and non-sphericity both over land and water. Overall, I think this paper is well-structured and clearly written, I recommend this paper to be published in AMT after some minor comments have been addressed.

**15 The authors thank reviewer 4 for their comments and recommendation.**

- 1. One interesting part however missing in the current manuscript is the direct comparison with MISR operational SA product. I would suggest to add at least some demonstrations of this part to show the evolution.
- Unfortunately, this paper (and our 2019 paper) are based on a subset of MISR data (Level 1 only) coincident with AERONET for a 4-year period. This dataset does not have MISR V23 data saved with it, which means making a direct comparison would be quite time-consuming. It would also make an already long manuscript even longer. We will absolutely do this in a future paper, as it is important for data users to understand the strengths and limitations of both algorithms (once the MISR research product is available to the public). Besides, such comparisons will be most relevant when we have completed our MISR Research *Product*, which we have recently been funded to produce, based largely on the work presented in this and other recent MISR RA papers.
  - 2. In the validation section, the authors evaluate the fine mode fraction with AERONET almucantar inversion product. Why not to use AERONET SDA FMF, which definitely will provide more coincidences?
- This is an excellent observation. The reason we used the almucantar inversions is to understand how MISR (and AERONET) SSA, FMF, and non-sphericity were impacting errors. For instance, we discovered that as AOD increases over water, coarse-mode non-sphericity was increasing. For a variety of reasons (we postulate in the manuscript), this resulted in poorer quality retrievals of FMF, SSA, and coarse-mode sphericity itself. It would not be possible to do this analysis with the AERONET SDA data, as we would be missing SSA and non-sphericity retrievals.

**#####SPECIFIC COMMENTS**

40

Page 4 Line 15: How the temporally interpolation is done? Meanwhile, how do you deal with the differences of MISR and MODIS wavelengths?

We have added that this is linear interpolation. Outside of the adjustments made to MAIAC surface reflectance (which we describe in the paper) we don't adjust for differences between MISR and MODIS spectral response, as these are small effects compared to other uncertainties.

Page 4 Line 20: MISR's 36 channels? This should be a mistake. Do you mean 4 wls x9 angles? 5 We have added that this means 4 bands by 9 cameras.

Page 5 Line 21: Surface reflectance correction? This is not clear to me. You correct your retrieved surface reflectance? If yes, how it can help to remove AOD bias?

- 10 or vou correct measured TOA reflectance? The prescribed surface retrieval initially produced even higher AOD biases than the ones presented in the paper. We used regions where the MISR retrieved surface algorithm agreed well with AERONET (and AOD was <0.2) to identify a correction for the MAIAC retrieved surface albedo (compared to the MISR retrieved surface albedo). We then applied this correction to all 9
- surface reflectances given by MAIAC for a specific band. Ideally, this would have eliminated the 15 prescribed surface bias. However, because we did not perform a camera-by-camera analysis, a significant bias remained in the data (unless the bias is calibration related). However, we only use the prescribed surface results when aerosol loading is high, so it is unlikely that a refinement in the prescribed surface reflectance will significantly impact our results.
- 20

Page 6 Line 15: How do you derive ANG from your algorithm, this is not clear in the text? ANG is derived by a log-log fit of wavelength to extinction using all 4 MISR bands. Extinction cross-section ratios (b.g.r.n/Green) are saved for all 17 MISR aerosol components, and effective extinction cross-section ratios are calculated based on the aggregate mixture fraction arising from a combination of all 17 components.

25

Page 14 Line 24: ANG at 550 nm?

This was a mistake. We have changed this to ANG (446-867 nm)

Page 15 Line 17: So the NDVI<0.1 is not retrieved over land, right? Or you still retrieve it but not pass 30 with high quality flag.

We still retrieve the data, but it did not pass our quality-assessment. We now have included retrievals with NDVI down to 0.0, which resulted in significantly more retrievals passing QA (including some desert retrievals)

Page 20 Line 1: it's not clear from Section 2. How the ANG is derived from the algorithm? Only AOD 35 at 550 nm is mentioned.

ANG is not initially retrieved by the algorithm, but instead calculated by the aerosol component fractions, as described in 2.1.2. The actual calculation of ANG is exactly as provided in our **AERONET** data and validation methodology.

40

Tables 4 and 6: it looks like incorrect for AOD blocks. 0.2<AOD<0.5 not 0.2>AOD>0.5? Thanks for catching that, you are correct.

**Reviewer 5**

5 This is a very good study describing research algorithm development for MISR. The standard MISR over-land retrieval has a long-standing problem of underestimating AOD at high AOD because the EOF algorithm fails when the surface contrast disappears at high AOD. This development uses prescribed MAIAC BRDF dataset over land (similar over ocean) to significantly improve the RA aerosol characterization at high AOD.

10

I recommend publication after the authors address my specific mostly editorial comments which I provide in the annotated manuscript. A minor re-structuring would also benefit this paper improving readability and understanding.

15 Alexei.

The authors thank Alexei Lyapustin for his comments and recommendation. We will add his comments with pages/line numbers (and our response) below.

20 Page 3, Line 2: The word "algorithm" makes it hard to understand - please remove. **Done.**

Page 3, Line3: Excessive, please remove.

25

Done.

Page 3, Line 8: Please, replace "or" with dash. **Done.**

Page 3, Line 18: Do you mean Lambertian water-leaving reflectance?

**30 We have changed this to remote-sensing reflectance.**

Page 3, Line 22: four to sixteen days, depending on latitude. MAIAC makes BRDF retrievals continuously, with every new orbit, but the BRDF is reported once in 8 days. In C6.1 it will be reported daily.

35 Clarified based on your comment.

Page 3, Line 27: I suggest to replace with "aerosol" for clarity. **Done**

40 Page 4, Line 21: MAIAC BRDF retrieval already uses similar constraints on BRDF shape to avoid unphysical behaviour of surface albedo with SZA (see 113. Lyapustin, A. I., Y. Wang, I. Laszlo, T. Hilker, F. Hall, P. Sellers, J. Tucker, S. Korkin, 2012. "Multi-angle implementation of atmospheric correction for MODIS (MAIAC): 3. Atmospheric correction." Rem. Sens. Env, 127: 385-393 [10.1016/j.rse.2012.09.002], see sec. "3.2. Solution selection and update" Added "(similar to constraints placed on MAIAC surface reflectances from *Lyapustin et al.*, 2012)."

5

Page 4, Line 26: You "consolidate the output ... to determine the surface type"? Please, re-write **We removed the redundancy from the end of the sentence.**

Page 5, Line 12: Can you come up with some other name, say "full retrieval", because "retrieved
surface aerosol retrieval" sounds very awkward and confusing, and in more than one place.
The authors have struggled with this as well. We used "retrieved surface aerosol retrieval"
because it was the best description of what is done. We have instead opted for "Retrieved Surface Algorithm" (RSA), "Prescribed Surface Algorithm" (PSA), and "Combined Surface Algorithm" (CSA).

15

Page 5, Line 16: I suggest to call it either "water leaving reflectance" or reflectance of underlight" which is a more appropriate and clear terminology for water. We changed this to remote-sensing reflectance.

20 Page 7, Line 14: I suggest that you say that this is a Lambert approximation of TOA (despite you are using the BRDF model, and use computed albedo, which, by the way, is also a function of AOD and SZA). But because your focus is high AOD retrievals, such approximation is reasonable and gives a good accuracy.

The authors admit this is an approximation and we add the following:

25 We recognize that this is only an approximation to account for multiple reflections of light off the surface.

Page 8, Line 21: I suggest to give this minimization equation first, prior to describing the specific set of equations (3) which follows from (4). This will make it more clear.

30 In this case, we prefer to keep the text as is.

Page 10, Line 3: You should mention that this is only an approximation. The water-leaving reflectance, theoretically, can be re-scattered by the water surface facets (we call glint) and the bulk of water after atmospheric backscattering, but the general formalism should be very different. In the ocean color

- 35 community, the "ocean color" component does not bounce between the atmosphere and the ocean and propagates directly to TOA. That means there is no denominator for A\*. The authors agree, but have handled this in a previous comment with the following: "We recognize that this is only an approximation to account for multiple reflections of light off the surface." The authors agree with your formalism, but in practical terms very little would
- 40 change as both s and A (especially A) tend to be very small numbers. We have added that A\* represents the remote-sensing reflectance over-water.

Page 11, Line 5: Please delete **Done.**

Page 11, Line 6: of **Done.**

Page 11, Line 19: Because you are using Lambertian formulation, you only retrieve a Lambertequivalent reflectance. The "angular behavior" of the surface reflectance usually implies the BRDF model information. The most recent analysis "Lyapustin A, Zhao F and Wang Y (2021) A Comparison

10 of Multi-Angle Implementation of Atmospheric Correction and MOD09 Daily Surface Reflectance Products From MODIS. Front. Remote Sens. 2:712093. doi: 10.3389/frsen.2021.712093"

clearly illustrates the differences between spectral BRF in MAIAC and Lambertian surface reflectance in standard MODIS surface reflectance MOD09.

15 The shape-similarity approximation (used over-land only) itself is a non-Lambertian surface reflectance model (whether accurate or not), although we are obviously using a Lambertian formalism for multiple reflections.

Page 11, Line 26: were

20 **Done.**

5

Page 13, Table 2, Row 1, Column 7: Personal comment, no action needed. You can reduce this dimension to 3-4 without change in accuracy.

Thanks for the comment, we will look into this in the future.

25

40

Page 14, Line 17: Do you mean 0.1-0.2? **Yes, changed.**

Page 15, Line 2: where AERONET "sphericity" is given in %.
30 We have removed this, as we now retrieve total sphericity (instead of CM sphericity).

Page 16, Line 32: I thought the algorithm is generic. Did you apply it for the deserts? It should work the same way for the dust retrievals. If it does not, you should say it upfront in the Abstract and Introduction/Conclusions.

35 We have now set the NDVI limit to 0.0, which will allow good QA retrievals over desert regions.

Page 16, Line 22: How is it defined? According to "2\*sigma, or 66% within EE"? This prognostic error is taken as a line fitted to the 68th percentile absolute AOD errors (with respect to AERONET), binned at every 2% of MISR retrieved AOD (so 50 bins in total). This has been added Page 17, Line 1: I don't quite get it: earlier you mentioned that you take "retrieved" when AOD<1, "prescribed" when AOD>2, and a superposition when AOD is in between. How ... No, it's possible, since you combine the best retrievals from each - no more questions. **Correct, we attempt to take the best pieces from each algorithm.**

5

Page 30, Line 5: That's the 1st place you mention what dataset was used in this paper. Plus, I need to gread the other paper before making my conclusions. This is important for understanding - I strong recommend that you add a small paragraph at the beginning explaining what regions (or global? - probably not because you filtered deserts), which years of MISR measurements are used here.

- 10 We have added the following in the MISR RA General Description: "The MISR top-ofatmosphere (TOA) reflectances used for this study are identical to the set of MISR reflectances used in our 2019 turbid water aerosol retrieval paper (Limbacher and Kahn, 2019), and represent 4 years of MISR data interspersed between 2001 and 2016 (over select direct-sun aerosol validation sites)." It is unlikely that all deserts are now filtered out, as we have lowered the NDVI
- 15 minimum to 0.0 and this should only filter out the most spectrally neutral (between the red and NIR) regions.